# *Bacteroides fragilis* polysaccharide A induces IL-10 secreting B and T cells that prevent viral encephalitis

Chandran Ramakrishna[1], Maciej Kujawski[1], Hiutung Chu[2], Lin Li[1], Sarkis K. Mazmanian[2] & Edouard M. Cantin [1]

The gut commensal *Bacteroides fragilis* or its capsular polysaccharide A (PSA) can prevent various peripheral and CNS sterile inflammatory disorders. Fatal herpes simplex encephalitis (HSE) results from immune pathology caused by uncontrolled invasion of the brainstem by inflammatory monocytes and neutrophils. Here we assess the immunomodulatory potential of PSA in HSE by infecting PSA or PBS treated 129S6 mice with HSV1, followed by delayed Acyclovir (ACV) treatment as often occurs in the clinical setting. Only PSA-treated mice survived, with dramatically reduced brainstem inflammation and altered cytokine and chemokine profiles. Importantly, PSA binding by B cells is essential for induction of regulatory CD4$^+$ and CD8$^+$ T cells secreting IL-10 to control innate inflammatory responses, consistent with the lack of PSA mediated protection in Rag$^{-/-}$, B cell- and IL-10-deficient mice. Our data reveal the translational potential of PSA as an immunomodulatory symbiosis factor to orchestrate robust protective anti-inflammatory responses during viral infections.

[1] Department of Molecular Immunology, Beckman Research Institute of City of Hope, Duarte, CA 91010, USA. [2] Division of Biology and Biological Sciences, California Institute of Technology, Pasadena, CA 91125, USA. Correspondence and requests for materials should be addressed to C.R. (email: rchandran@coh.org) or to E.M.C. (email: ecantin@coh.org)

Mammals are colonized by an astounding number of diverse microorganisms[1,2], collectively referred to as the microbiota, whose gene content exceeds that of the host by 100-fold[3]. Bacterial species belonging to the phyla Bacteroidetes and Firmicutes dominate the microbiota. Disruption of the gut microbiota (dysbiosis) characterizes a variety of chronic inflammatory diseases, including inflammatory bowel disease (IBD), arthritis, and multiple sclerosis (MS)[4,5]. Several bacterial species with immunomodulatory activity have been identified and shown to have the capacity to correct gut dysbiosis, thereby alleviating diverse inflammatory diseases[6–8]. *Bacteroides fragilis* (*B. fragilis*) and its immunomodulatory capsular polysaccharide A (PSA)[9], have been extensively studied and shown to be equally effective in preventing colitis and experimental allergic encephalomyelitis (EAE) in murine models[10–12]. The immunomodulatory activities of PSA induce regulatory T cells secreting IL-10, a potent anti-inflammatory cytokine that restrains pathogenic inflammation in the gut, as well as systemically including in the brain[12–15]. However, PSA has only been tested for its ability to ameliorate slowly progressing sterile inflammatory diseases.

There are reports that the gut microbiota can influence the outcome of certain viral infections including influenza virus, coxsackie virus, and Friend leukemia virus, as germ-free (GF) mice were more susceptible than specific pathogen-free (SPF) mice (reviewed in refs. [16–18]). Other studies revealed that the microbiota were required for adaptive immune responses to murine cytomegalovirus (MCMV), a homolog of human CMV[19,20], a β-herpesvirus that causes inflammatory diseases, such as gastritis and pneumonitis in immunosuppressed individuals[17]. Although the use of GF mice supports a link between the microbiota and adaptive immune responses to MCMV infection, a caveat is that lymphoid organ development and immune responses are impaired in GF animals[21,22]. A recent study reported that oral antibiotic treatment caused dysbiosis of the vaginal microbiota, resulting in increased IL-33 secretion, leading to impaired antiviral T cell responses and enhanced susceptibility to lethal HSV2 infection[23].

Globally, HSV1 is the leading cause of sporadic encephalitis, which despite improvements in antiviral treatment, is still associated with high mortality, debilitating neurological outcomes, and a greatly impaired quality of life for survivors. We have shown that fatal herpes simplex encephalitis (HSE) in susceptible 129 mice results from immune pathology rather than virus replication-induced damage[24,25]. The standard of care antiviral, ACV, protects when administered early (≤day 2 pi) but not at later times in infection (day 4 pi), because although virus replication in the brain is rapidly suppressed, inflammation continues to escalate, culminating in fatal HSE. Notably, treatment with intravenous immunoglobulins (IVIG) that possess potent immunomodulatory activity protected against fatal HSE by inducing regulatory T cells secreting IL-10, consistent with HSE being a neuroinflammatory disease[26,27]. To expand applicability of PSA beyond sterile inflammatory disease models, we investigated its ability to protect against HSE. Naïve mice pretreated with PSA survived a lethal HSV challenge, whereas PBS-pretreated mice succumbed, despite treatment of all mice with ACV. PSA reduced brainstem (BS) inflammation, altered cytokine and chemokine profiles, induced IL-10 secreting ICOS+ CD39+CD73+CD4+ T cells and virus-specific CD73+CD8+ T cells. A novel finding is that gut resident B-cells bound PSA and B cells were crucial for induction of regulatory T cells secreting IL-10 that are mandatory for restraining pathogenic innate inflammatory responses in the CNS. We conclude that PSA utilizes distinct pathways to prevent sterile inflammatory diseases, such as EAE, compared to optimizing immunity in naïve mice to render them resistant to a subsequent lethal viral infection. These

data demonstrate that a bacterial immunomodulatory symbiosis factor can optimize host immunity, encouraging further exploration of combinatorial antiviral–bacterial immunomodulatory factor or nutritional approaches for treating viral-induced inflammatory diseases.

## Results

**Early but not delayed treatment with acyclovir prevents HSE.** Several mouse studies have reported that *B. fragilis* or purified PSA can prevent various sterile inflammatory diseases by inhibiting pathogenic inflammatory cells in the gut as well as in the brain and lung[11,12,28,29]. However, whether probiotic treatment can be beneficial in virus-induced inflammatory diseases is unknown. To address this question, we assessed the immunomodulatory potential of *B. fragilis* and PSA in a murine model of HSE. We have previously shown that HSE results from unrestrained CNS inflammation[24–26]. ACV, the standard of care antiviral drug, is protective when given early (≤day 2 pi), but its efficacy declines rapidly when treatment is delayed (Supplementary Fig. 1a). Survival plummets to 25% when ACV is given from day 4 pi because despite efficient inhibition of virus replication by day 6 pi, CNS inflammation escalates unimpeded culminating in fatal HSE[25,30]. We gave ACV from day 4 pi in our studies, as this regimen effectively separates the effects of virus replication and inflammation on development of fatal HSE, enabling studies focused on the immunomodulatory effects of PSA in protection against HSE.

**Oral treatment with PSA protects against viral encephalitis.** We first administered PSA to HSV infected 129 mice on days 1, 2, and 4 pi via the intraperitoneal (ip) or intravenous (iv) routes or by oral gavage and treated them with ACV from day 4 pi. All mice succumbed to HSE (Supplementary Fig. 1b). Since, HSE is a rapidly evolving neuroinflammatory disease, we next determined whether PSA pre-treatment prior to challenge with HSV could protect mice from HSE. Six doses of PSA, but not PBS, administered by oral gavage, but not via the ip or iv routes, over a span of 21 days before HSV infection protected the majority of mice from fatal HSE (Fig. 1 and Supplementary Fig. 1c). PSA given prior to infection but without ACV treatment was not protective (Supplementary Fig. 1d). Thus, our experimental approach for all subsequent experiments was to treat mice with six doses of PSA (50 μg) by oral gavage over 3 weeks, followed by infection with HSV and ACV given daily from day 4 pi for a week (Fig. 1a). We also evaluated *B. fragilis* delivered by oral gavage prior to challenge with HSV and ACV treatment according to the scheme in Fig. 1a. As expected, *B. fragilis* protected against HSE as effectively as PSA (Fig. 1b). PSA was unable to protect Rag−/− mice from HSE (Fig. 1b), which suggested that either T cells or B cells or both cell subsets are required for PSA's anti-inflammatory mechanism.

**PSA restrains inflammatory myeloid cells to prevent HSE.** We reported that HSE results from unrestrained infiltration of the BS by CD45high leukocytes, specifically Ly6Chigh inflammatory monocytes (IM) and Ly6G+ neutrophils[25,26]. Analysis of the BS mononuclear cells revealed that PSA suppressed infiltration of CD45high leukocytes into the BS of wildtype (WT) but not Rag−/− mice (Fig. 2). The gating strategy to determine various infiltrating leukocyte subsets in the BS is shown in Supplementary Fig. 2a. The percentages and absolute numbers of CD45high leukocytes and Ly6Chigh IM infiltrating the BS of PSA and PBS treated Rag−/− mice were similar (Fig. 2a, b). In contrast, leukocyte infiltration was dramatically reduced in the BS of PSA treated compared to PBS-treated WT mice. While CD45high

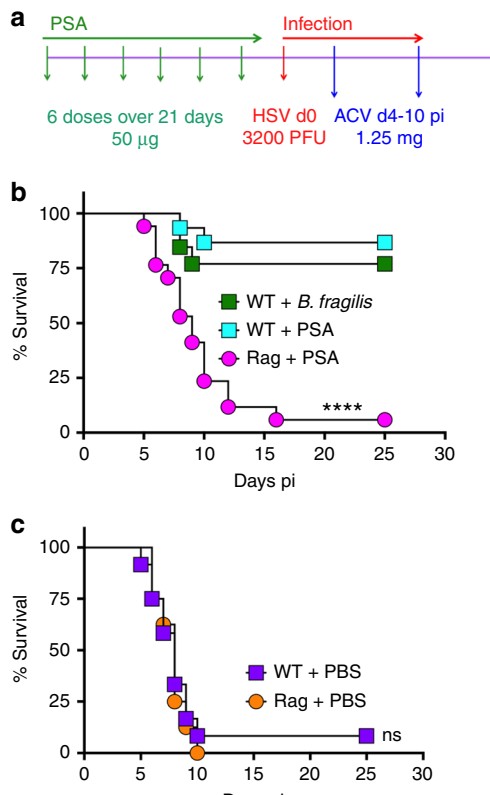

**Fig. 1** *B. fragilis* PSA protects against HSE. **a** Experimental regimen: In all experiments, PSA (six doses, 50 µg/mouse) or PBS was given orally before HSV infection on day 0 and thereafter daily ip injections of ACV from day 4 pi for 7 days. Survival of wildtype (WT) or Rag mice pre-treated with **b** *B. fragilis*, PSA or **c** PBS, (n = 8–17 mice/group). ****p < 0.001, WT + *B. fragilis*/PSA vs. Rag + PSA; ns: not significant, WT + PBS vs. Rag + PBS as determined by log rank (Mantel–Cox) test

infiltrates comprised ~75% of BS cells in PBS-treated WT mice, this population declined to ~15% in the BS of PSA-treated WT mice, representing an ~20-fold reduction in numbers of infiltrating cells (Fig. 2c). This was reflected by a substantial reduction of CD11b$^+$ cells including monocytes and macrophages and Ly6G$^+$ neutrophils, and importantly, a marked reduction of degranulating IM in the BS of PSA-treated mice (Fig. 2d, e). In contrast, PSA treatment increased the accumulation of T cells, especially CD8 T cells in the BS (Fig. 2f).

**PSA induces regulatory T cell populations**. Previously, PSA was shown to induce or activate FoxP3$^+$CD4$^+$ Tregs that are required for protection in various models of autoimmune inflammatory diseases[11,13,28,29]. Therefore, we investigated whether FoxP3$^+$ Tregs were similarly induced following HSV infection. As expected, analysis of spleen and draining cervical lymph nodes (CLN) revealed an almost two-fold increase of FoxP3$^+$ Tregs in spleens of PSA treated compared to PBS-treated mice (Fig. 3a, c). Importantly, PSA-induced FoxP3$^+$ Tregs expressed high levels of CD25 and CD103 surface markers indicative of a potent functional phenotype, whereas effector CD4 T cells tended to express lower levels of CD69 (Fig. 3b–f). Unexpectedly, depletion of Tregs did not eliminate protection against HSE, suggesting that PSA induces multiple redundant compensatory mechanisms to protect against HSE (Fig. 3g).

Remarkably, administration of six doses of PSA to uninfected WT mice by oral gavage induced substantial increases in CD3$^+$CD4$^+$ and CD3$^+$CD8$^+$ T cells in the draining mesenteric

lymph nodes (MLN) early in the course of PSA treatment (Fig. 3h, i); the gating strategy is shown in Supplementary Fig. 2b. Notably, accumulation of CD4 and CD8 T cells expressing regulatory CD39 and CD73 molecules increased progressively with PSA treatment (Fig. 3j and Supplementary Fig. 2c). PSA-induced regulatory CD39$^+$CD4$^+$ T cells were previously reported to protect from EAE[31]. Intriguingly, there were more CD73$^+$ compared to CD39$^+$ CD4 and CD8 T cells at the end of PSA treatment (Fig. 3j). Additionally, a significant increase in CD4 T cells expressing ICOS, a potent CD4 T cell activation marker was also observed (Fig. 3j)[26]. Following virus infection, increased numbers, but similar percentages of virus-specific CD8 T cells were detected in the draining CLN of PSA treated, compared to PBS-treated WT mice (Supplementary Fig. 3a, e). Additionally, higher numbers and percentages of antigen-specific CD4 T cells, depicted by CD11a expression, were present in PSA-treated infected mice and they also expressed increased levels of CD73 (Supplementary Fig. 3b, e). Interestingly, virus-specific CD8 T cells and plasma cells (PC) were increased in the lamina propria (LP) of PSA-treated mice (Supplementary Fig. 3c, d).

**IL-10 and IFNγ are required for PSA protection against HSE**. To elucidate the mechanism(s) induced by PSA that protect from HSE, we analyzed cytokines and chemokines in BS obtained from HSV-infected WT mice pretreated with PSA or PBS. Interestingly, PSA treatment increased IL-10 and IFNγ expression and of these cytokines, IL-10 is a known anti-inflammatory mediator (Supplementary Fig. 4a). Although considered a pro-inflammatory cytokine, we determined recently that IFNγ suppresses G-CSF-induced neutrophilia and thereby prevents development of fatal HSE[27]. Unsurprisingly then, expression of the IFNγ inducible chemokines CXCL10 and CXCL11 was at much higher levels in BS of PSA treated compared to PBS-treated mice, while increased expression of CCL4, a Th2-related chemokine was observed in PBS-treated WT mice (Supplementary Fig. 4b). Additionally, PBS-treated WT mice expressed higher levels of pro-inflammatory cytokines and chemokines including CCL2, CCL7, and IL-6, which induce production and emigration of inflammatory neutrophils and IM from the BM into the BS. Since we previously established that IL-10 and IFNγ, respectively, control expansion of IM and neutrophils during HSE, we next explored whether PSA induces IL-10 and IFNγ production. Both CD4 and CD8 T cells isolated from MLN of PSA-treated infected, but not uninfected WT mice, produced IL-10 and IFNγ as revealed by intracellular flow cytometry (Fig. 4a, b and Supplementary Fig. 4c, d). Thus, although PSA treatment increases regulatory CD73$^+$ and CD39$^+$ CD4 and CD8 T cells (Fig. 3j), these cells make IL-10 and IFNγ only after HSV infection. Since PSA treatment also induced accumulation of B cells in the MLN, we analyzed them for IL-10 and IFNγ and found surprisingly, that B cells also secreted IL-10 and IFNγ (Fig. 4a, b and Supplementary Fig. 4c, d).

Next, we investigated whether both IL-10 and IFNγ were necessary for PSA protection against fatal HSE. IL-10 and IFNγ knockout (IL-10KO and GKO) mice pretreated with six doses of PSA or PBS according to the scheme in Fig. 1a were challenged with HSV and given ACV from day 4 pi. Unexpectedly, PSA protection was impaired in both IL-10KO and GKO mice that all succumbed to HSE (Fig. 4c, d) because PSA failed to suppress inflammation in the BS (Fig. 4e, f). Elevated numbers of CD45$^{high}$ leukocytes present in the BS of these mice were characterized by increased accumulation of IM in BS of IL-10KO mice and Ly6G$^+$ neutrophils (PMN) in the BS of IFNγKO mice (Fig. 4e, f). Thus, both IL-10 and IFNγ are required to suppress accumulation of IM and neutrophils, respectively, in the BS, as we have shown previously[26,27].

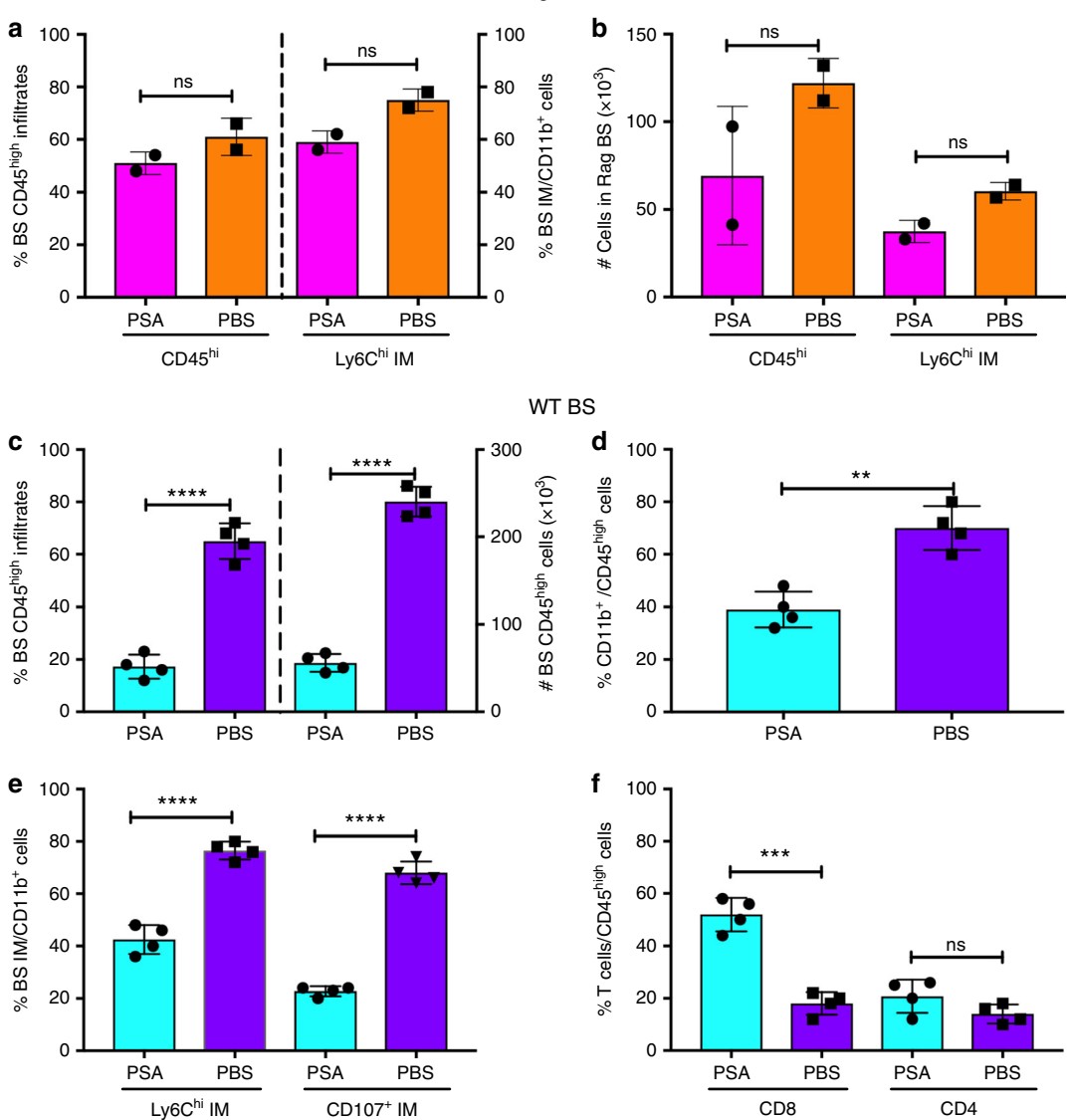

**Fig. 2** PSA reduces CNS inflammation in HSV-infected WT but not Rag mice. **a** % and **b** total numbers (#) of CD45$^{high}$ leukocytes and CD45$^{high}$ Ly6C$^{high}$ inflammatory monocytes (IM) infiltrating the brainstem (BS) of Rag mice. **c** % (left y-axis) and # (right y-axis) infiltrating CD45$^{high}$ leukocytes in the BS of WT mice. **d** % CD11b$^+$ cells within BS infiltrating CD45$^{high}$ cells; **e** % Ly6C$^{high}$ and CD107$^+$ IM within the CD11b$^+$ population; **f** % CD4$^+$ and CD8$^+$ T cells within CD45$^{high}$ cells in the BS of WT mice. Data compiled from 2 to 4 experiments with $n = 6$–$8$/group at day 6 pi. All data show mean ± SEM. ***$p <$ 0.0005, ****$p <$ 0.0001, ns: not significant, as determined by two-tailed Student's $t$-test

The inability of PSA to protect Rag$^{-/-}$ mice from HSE, suggests that either T cells or B cells or both lymphocyte subsets are mandatory for PSA-mediated protection (Fig. 1b). Furthermore, both T and B cells in the MLN, CLN, and spleens of PSA-treated HSV-infected WT mice produced IL-10 and IFNγ (Fig. 4a, b). However, only CD8 T cells and a few CD4 T cells, but not B cells, were detected in the BS of PSA-treated infected WT mice (Fig. 2f). To determine which cell subset is necessary for PSA-mediated protection from HSE, we transferred WT CD4 or CD8 T cells or B cells into groups of Rag$^{-/-}$ mice one week prior to treatment with PSA as depicted schematically in Fig. 5a (in black text). Recipient Rag$^{-/-}$ mice were treated with six doses of PSA or PBS before challenging them with HSV and administering ACV. Unexpectedly, all three groups of Rag recipients pretreated with either PSA or PBS succumbed to HSE (Fig. 5b and Supplementary Fig. 1e). Furthermore, anti-mouse CD20 antibody-mediated depletion of B cells in WT mice (BKO) abrogated PSA protection (Fig. 5b and Supplementary Fig. 5a),

which confirms that B are mandatory for protection from fatal HSE. Interestingly, B cell depletion reduced T cell accumulation in the draining CLN at day 6 pi (Supplementary Fig. 3e) and surprisingly abrogated IL-10, but not IFNγ secretion by CD4 and CD8 T cells in PSA-treated infected BKO mice (Supplementary Fig. 5b). These results suggest that B cells are required for PSA to promote IL-10 secretion by T cells.

**IL-10 from B and T cells is required to prevent fatal HSE.** To determine if the source of IL-10 was important for PSA protection from HSE, we performed mixed transfers of WT and IL-10KO T and B cells into Rag$^{-/-}$ recipients. One group of Rag$^{-/-}$ recipients received WT CD3$^+$ T cells and IL-10KO CD19$^+$ B cells (brown), while the second group received IL-10KO CD3$^+$ T cells and WT CD19$^+$ B cells (green). The control Rag$^{-/-}$ recipients received either IL-10KO B and T cells (magenta) or WT B and T cells (blue) to recreate IL-10KO or WT mice, respectively. All

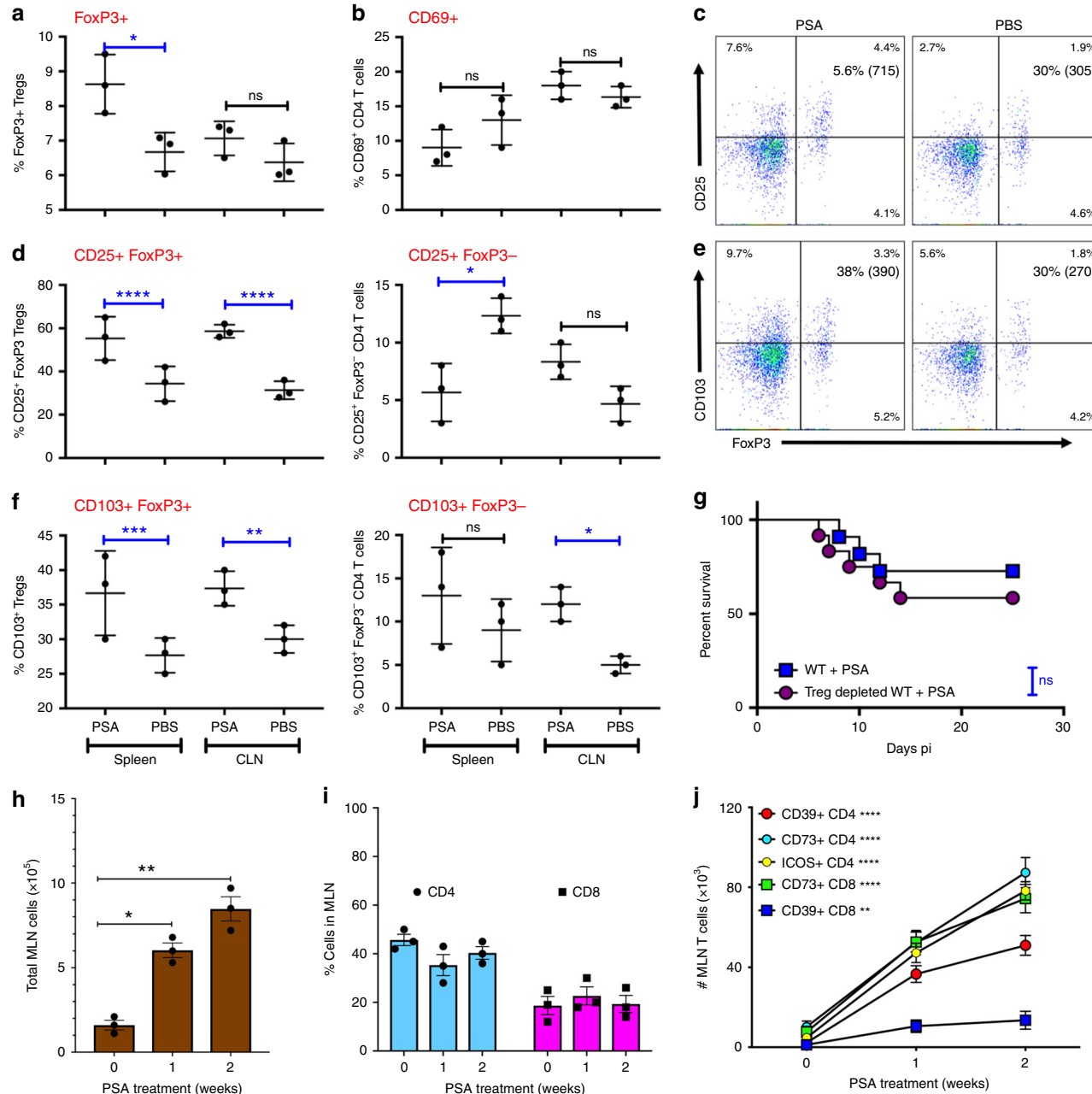

**Fig. 3** PSA protection from HSE is independent of induced Tregs. **a** % FoxP3+ CD4 Tregs and **b** CD69+ CD4 T cells in spleen and CLN of PSA or PBS-treated WT mice at day 6 pi. **c** CD25 expression within FoxP3+ CD4 Tregs in WT mice at day 6 pi, % and mean fluorescence intensity (MFI) in () shown in right top quadrant. **d** % CD25 within FoxP3+ Tregs (left plot) and FoxP3− CD4+ T cells (right plot), **e** CD103 expression within FoxP3+ Tregs in WT mice at day 6 pi; % and MFI in () shown in right top quadrant. **f** CD103 within FoxP3+ Tregs (left plot) and FoxP3− CD4+ T cells (right plot) in the spleen or CLN of WT mice at day 6 pi. Data from three experiments shown. **g** PSA-treated Treg depleted and control WT mice were monitored for survival after HSV infection and ACV treatment as in Fig. 1a, ns: not significant determined by log rank Mantel−Cox test ($n = 11$–12 mice). After administration of three (1 week) or six doses (2 weeks) of PSA, MLN in uninfected WT mice were monitored for **h** cellularity, **i** % CD4 and CD8 T cells, and **j** # ICOS+, CD39+, and CD73+ CD4 and CD8 T cells ($n = 3$ mice); ****$p < 0.0001$, **$p < 0.01$ as determined by two-way ANOVA or one-way ANOVA with Sidaks or Turkeys correction, respectively, for multiple comparisons tests. All data show mean ± SEM

groups of Rag$^{-/-}$ recipients were treated with six doses of PSA or PBS, infected with HSV and then treated with ACV according to the scheme in Fig. 5a. Recapitulating results obtained with WT and IL-10KO mice, PSA but not PBS-treated Rag$^{-/-}$ recipients of WT B and T cells were protected from subsequent HSV challenge, while Rag$^{-/-}$ recipients of IL-10KO B and T cells succumbed to fatal HSE (Figs. 1b, 4c, 5c and Supplementary Fig. 1e). Most of the mixed transfer recipients of WT B cell and IL-10KO T cell subsets also succumbed to HSE. Although, protection of

Rag$^{-/-}$ recipients of IL10KO B cell and WT T cells was augmented (~50%), this was not statistically significant (Fig. 5c). Analysis of BS infiltrates in all four groups of mice revealed that only Rag$^{-/-}$ mice that received WT T and B cells had reduced CD45$^{high}$ infiltrates in the BS, whereas all other groups of Rag$^{-/-}$ recipients exhibited high levels of inflammation characterized by CD45$^{high}$ cells comprised predominantly of either CD11b+ CCR2+ Ly6C$^{high}$ IM or CD11b+Ly6G+ PMN (Fig. 5d, e and Supplementary Fig. 5c). Since Rag recipients of donor WT or

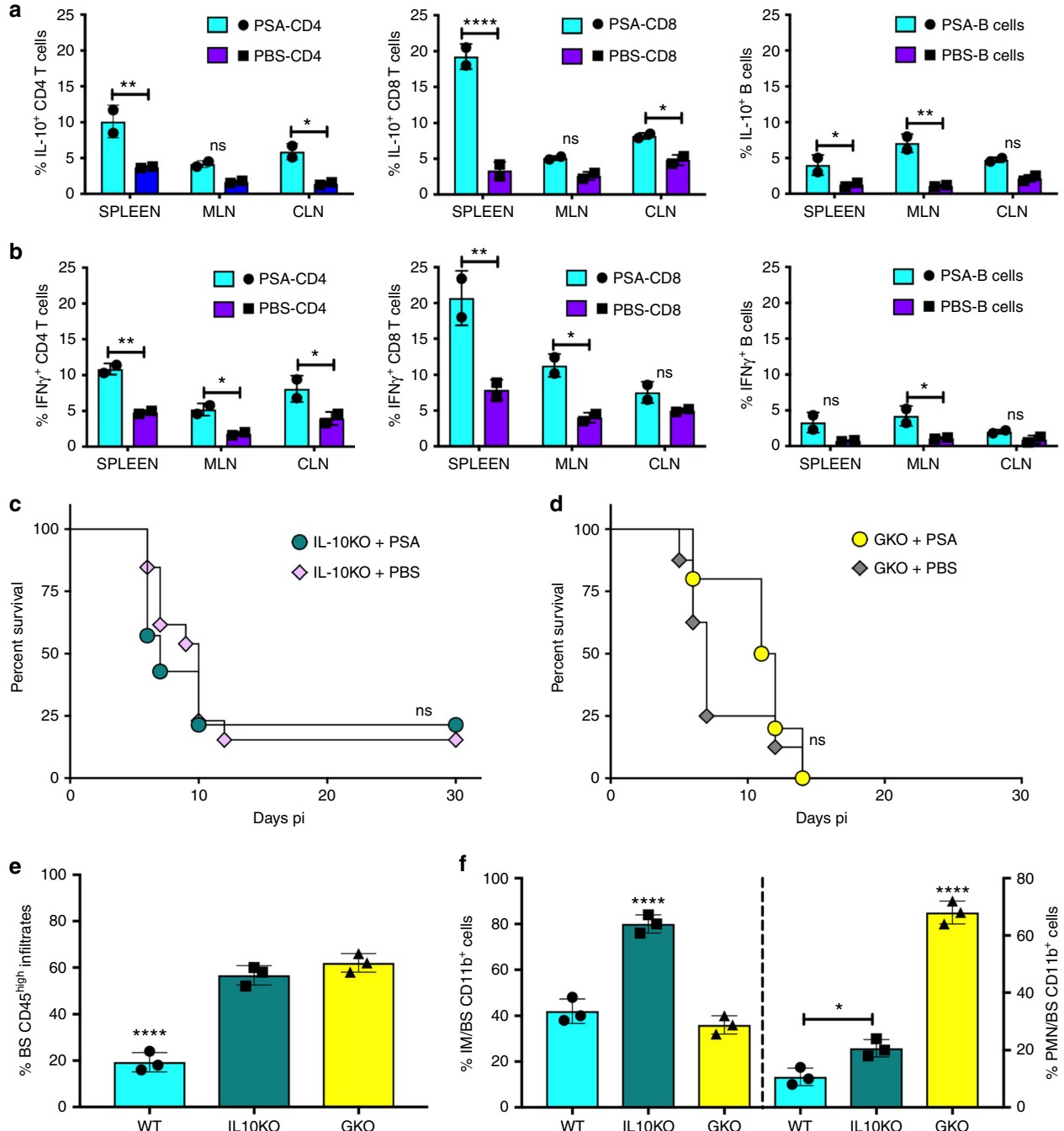

**Fig. 4** PSA increases IL-10 and IFNγ-secreting T cells. CD4 and CD8 T cells and B cells in spleens, mesenteric lymph nodes (MLN), and cervical lymph nodes (CLN) of PSA or PBS-treated WT mice at day 6 pi were analyzed for **a** IL-10 and **b** IFNγ secretion, $n = 2$ experiments; $*p < 0.05$, $**p < 0.01$, $****p < 0.0001$, as determined by two-way ANOVA with Sidak's multiple comparisons test. Survival of PSA or PBS treated **c** IL-10KO mice or **d** IFN-GKO mice ($n = 8$–16 mice); ns: not significant. Bar plots show **e** % CD45$^{high}$ leukocytes, **f** (left $y$-axis) % Ly6C$^{high}$ IM and (right $y$-axis) % Ly6G$^+$ neutrophils (PMN) within CD45$^{high}$ CD11b$^+$ cells infiltrating the BS of PSA treated 129 WT, IL10KO, and GKO mice at day 6 pi, $n = 3$ experiments with 2–3 BS/group; $*p < 0.05$, $****p < 0.0001$ as determined by ordinary one-way ANOVA with Turkey's multiple comparisons tests

IL10KO lymphocytes had similar proportions of CD45$^+$ leukocyte subsets infiltrating the BS of WT and IL10KO mice, respectively, these data are combined and shown in Fig. 4e, f. Intriguingly, distinct inflammatory infiltrates were observed in recipients of the two mixed transfer groups (Fig. 5d, e). Rag$^{-/-}$ recipients of WT B cells and IL-10KO T cells (green) had ~68% CD45$^{high}$ infiltrates in the BS compared to ~51% infiltrating CD45$^{high}$ leukocytes for recipients of (brown) IL-10KO B cells and WT T cells (Fig. 5d, e). The majority of CD45$^{high}$ infiltrates

in the BS of the IL-10KO B cell + WT T cell recipients were neutrophils (CD45$^{high}$ CD11b$^+$ infiltrates: 91% PMN, 5% IM, Fig. 5d). In contrast, infiltrates in BS of the WT B cell + IL-10KO T cell recipients were dominated by Ly6C$^{high}$ IM (CD45$^{high}$ CD11b$^+$ cell infiltrates: 72% IM, 16% PMN, Fig. 5e). And, while similar levels of HSV gB$_{498-505}$-specific CD8 T cells and ICOS$^+$CD4$^+$ T cells were observed in both mixed transfer groups, they expressed a highly activated (CD44$^{high}$) effector T cell phenotype in the spleen (Supplementary Fig. 5d). These

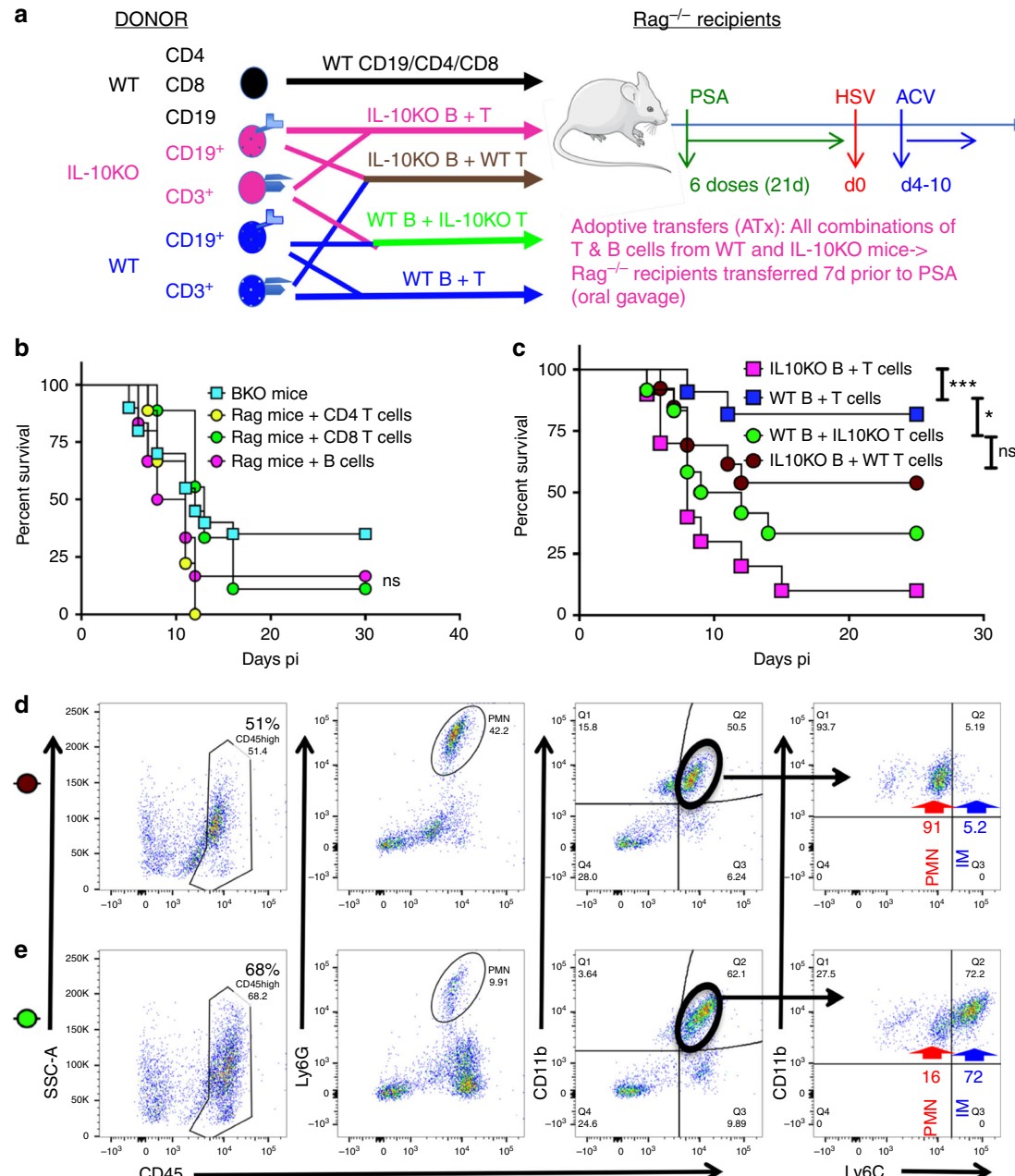

**Fig. 5** PSA protection against HSE requires B and T cells secreting IL-10. **a** Experimental design for experiments in **b** and **c** Donor WT (In black text): Naïve Rag mice were transferred with WT CD4+ or CD8+ T cells or CD19+ B cells 7 days before PSA treatment. Donor IL-10KO (magenta text) and WT (Blue text): four groups of naïve Rag mice were transferred with combinations of donor WT B and T cells, IL-10KO B and T cells, WT B and IL-10KO T cells, IL-10KO B and WT T cells 7-days before PSA treatment. All Rag recipients received six doses of PSA before HSV infection and ACV treatment. **b** Survival of B cell-depleted mice (BKO, $n = 20$ mice) and Rag recipients of WT single cell subsets ($n = 6$–9 mice/group). B cell depletion in WT mice was initiated 10 days prior to PSA treatment and continued throughout infection, ns: not significant. **c** Survival of Rag recipients of WT and IL-10KO combination of T and B cells ($n = 10$–13/group). ***$p < 0.001$, *$p < 0.05$, ns: not significant as determined by log rank (Mantel–Cox) test. FACS plots of BS CD45$^{high}$ cells (left), Ly6G+ PMN (left middle), CD11b+ cells within CD45$^{high}$ cells (right middle), and Ly6C$^{high}$ IM and Ly6C$^{int}$ CD11b+ PMN within CD45$^{high}$ CD11b+ cells (right) were analyzed at day 6 pi in the BS of Rag recipients of **d** IL-10KO B + WT T cells (brown circle) and **e** WT B + IL-10KO T cells (green circle)

results show that IL-10 has distinct effects on the composition of inflammatory infiltrates in the BS depending on whether T or B cells are the source.

**Intestinal plasmacytoid DCs and B cell plasmablasts (PB) bind PS.** Plasmacytoid dendritic cells (pDC) and macrophages were reported to bind PSA and induce IL-10-secreting regulatory T cells[9,32]. We investigated whether the binding of PSA to disparate immune cell populations in the intestines of WT and

Rag$^{-/-}$ mice accounted for the divergent outcomes of HSV infection in these strains. Sections of the intestine including duodenum, ileum, and colon were isolated from WT and Rag$^{-/-}$ mice that had been orally gavaged with fluorescently conjugated PSA. CD45 was used as a marker to discriminate cells of hematopoietic origin (CD45+) from gut (CD45−) epithelial cells (Fig. 6a). CD45− intra-epithelial cells (IEC) in both groups of mice showed binding to PSA. Although, CD45+ intra-epithelial leukocytes (IEL) in the duodenum and ileum of Rag$^{-/-}$ mice did

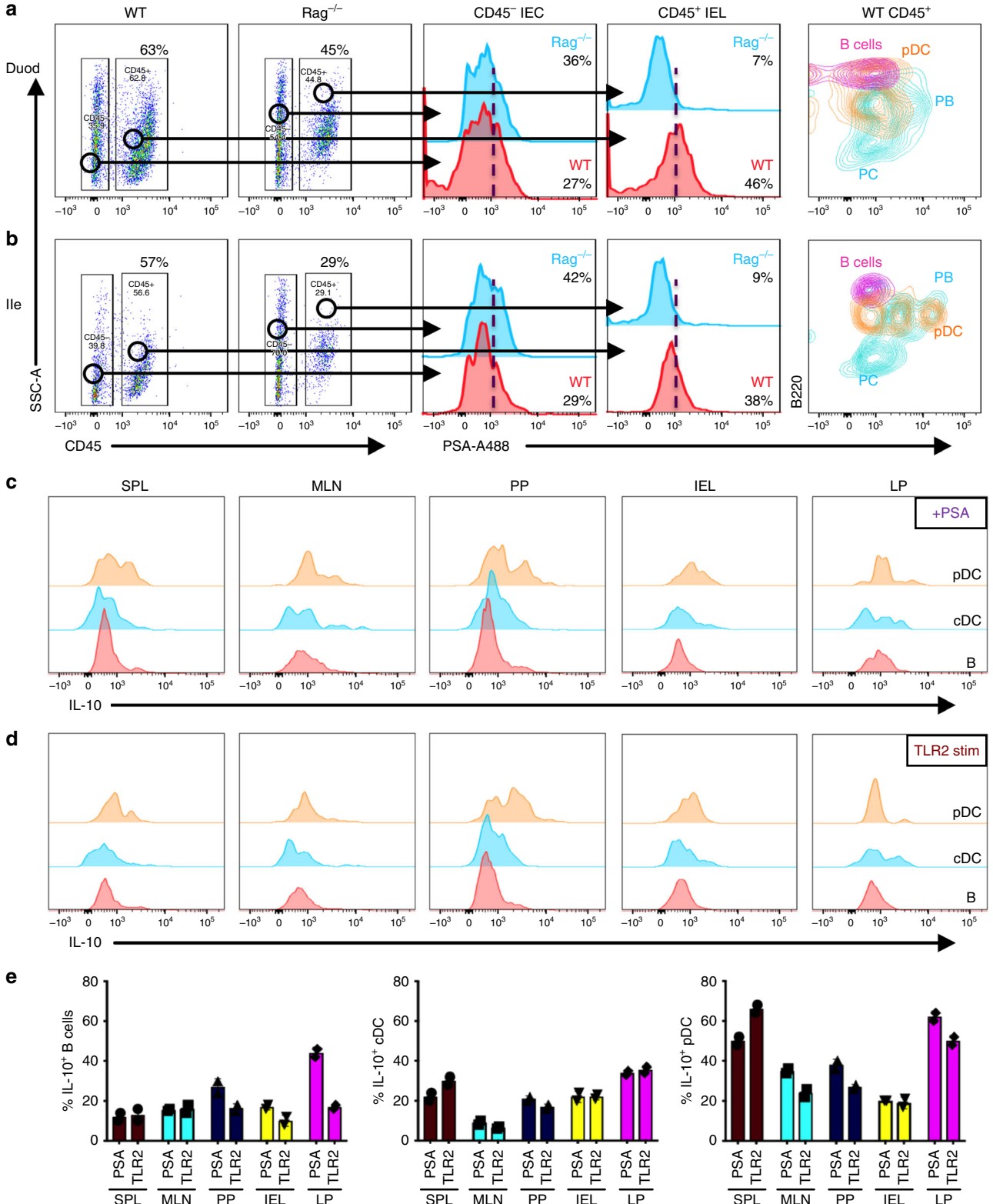

**Fig. 6** TLR2[+] macrophages, pDCs, and PB in the small intestine bind PSA and induce IL-10 secretion. Gating strategy for mononuclear cells isolated from **a** duodenum (Duod) and **b** ileum (Ile) of WT and Rag mice analyzed for binding of fluorescent A488-conjugated PSA (left two plots). CD45[−] intra-epithelial cells (CD45[−] IEC: middle histogram) and CD45[+] gated intra-epithelial leukocytes (CD45[+] IEL: second right histogram) isolated from the **a** Duod and **b** Ile of WT (red) and Rag (blue) mice were analyzed for reactivity to PSA-A488. CD45[+]CD11c[−]B220[+] B cells, PDCA1[+] B220[+]CD11c[+] pDCs and CD138[+]B220[+] PB and B220[low] PC isolated from **a** Duod and **b** Ile of WT mice were analyzed for PSA reactivity (right histogram). Flow cytometry plots show CD11c[+]PDCA1[−] cDC, PDCA1[+]B220[+] pDC, and B220[+]CD19[+] B cells isolated from spleen, MLN, PP, IEL, or LP of WT mice were stimulated with **c** PSA or **d** LTA-SA (TLR2 agonist) and analyzed for IL-10 expression. **e** Plots summarize data from **c** and **d** and show % IL-10[+] B cells (left), cDCs (middle), and pDCs (right) from spleen, MLN, PP, IEL, and LP stimulated with PSA or TLR2 (LTA-SA) (n = 3 mice). All data show mean ± SEM

not bind PSA, CD45$^+$ cells in all sections of ileum and colon of WT mice bound PSA (Fig. 6a, b). Duodenal and ileal B220$^+$PDCA1$^+$CD11c$^+$ pDCs and B220$^+$CD138$^+$ PB, but not B220$^{low}$CD138$^+$ PC or B220$^+$CD19$^+$ B cells, bound PSA with the highest intensity (Fig. 6a, b and Supplementary Fig. 6a–d). Moreover, PSA-treated WT duodenal CD45$^+$ IELs secreted higher levels of IL-10 and reduced levels of IL-17A whereas in contrast, PSA treated Rag$^{-/-}$ duodenal CD45$^+$ IELs secreted increased IL-17A but not IL-10 (Supplmentary Fig. 6e, f). Similarly, cells isolated from ileum and Peyer's patches (PP) of Rag$^{-/-}$ mice showed increased IL-17A production after stimulation with PSA or PMA and Ionomycin (Supplementary Fig. 6g). The binding to PSA was specific because prior incubation of spleen cells with cold unlabeled PSA inhibited binding to fluorescently labeled PSA in vitro (Supplementary Fig. 7a). Similar to the binding observed in the gut, PB, pDCs, and a subset of F480$^+$ macrophages but not T cells or Gr-1$^+$Ly6G$^+$ neutrophils (PMN) bound PSA in the spleen (Supplementary Fig. 7b–h).

PSA binding to immune cells via TLR2 was shown previously to be essential for induction of its immunomodulatory effects[32]. We analyzed TLR2 expression on gut resident cells and splenocytes which bound PSA to determine if TLR2-expressing cells was essential for protection against HSE (Supplementary Fig. 8). As expected, TLR2 was expressed on pDCs, cDCs, macrophages, and a subset of B cells including PB and, these cells also bound PSA with pDCs showing the strongest binding (Supplementary Fig. 8a). Interestingly, a subset of CD45$^-$ IECs that expressed TLR2 also showed strong binding to PSA, leading to speculation that they may play an important role in initial binding of PSA. Further, splenic DC subsets and F480$^+$ macrophages expressed high levels of TLR2 and TLR4 and bound PSA (Supplementary Figs. 8b, 7g), Strangely, PMNs did not bind PSA despite expressing high levels of TLR2 and TLR4 (Supplementary Figs. 7h, 8b). To ascertain whether TLR2 was essential for induction of IL-10, we stimulated pDCs, cDCs, and B cells isolated from spleen, MLN, peyers patches (PP), IEL, and LP with PSA or a TLR2-specific agonist LTA-SA (Fig. 6c–e). B cells from LP and PP stimulated with PSA but not LTA-SA secreted IL-10, whereas B cells from spleen, MLN, and IEL did not. Both cDCs and pDCs from spleen, PP, IEL, and LP responded to both PSA and TLR2 stimulation by making IL-10 (Fig. 6c–e). Although pDCs bound PSA robustly and secreted IL-10, anti-PDCA1 mediated depletion of pDCs in PSA-treated infected mice did not abrogate protection from HSE, suggesting that pDCs play a redundant role in protection from HSE (Supplementary Fig. 9a).

Since the C-type lectin receptor SIGNR1 is co-expressed with Toll-like receptors on pDCs and a subset of macrophages, and is a known receptor for various microbial polysaccharides, we next investigated if cells from duodenum and ileum of SIGNR1-deficient (SIGNR1KO) mice bound PSA. CD45$^+$ IELs, but not CD45$^-$ IECs from duodenal and ileal segments of SIGNR1KO mice showed reduced binding to PSA compared to WT mice (Supplementary Fig. 9b). Importantly, pDCs and macrophages, but not PB isolated from duodenum of SIGNR1KO mice showed reduced binding to PSA (Supplementary Fig. 9c, d). Nevertheless, pretreatment with PSA but not PBS according to the scheme in Fig. 1a rendered these mice resistant to HSE (Supplementary Fig. 9e). These results suggest that SIGNR1 plays a limited or redundant role in PSA-mediated protection.

Administration of six doses of PSA progressively increased accumulation of PC and to a lesser extent PB in MLN and spleens of WT mice (Supplementary Fig. 10a). Following HSV infection of Rag$^{-/-}$ recipients of the mixed WT and IL10KO T and B cell transfer groups shown in Fig. 5a, the absence of IL-10 secretion in the spleens of Rag$^{-/-}$ recipients of IL-10KO T cells was surprisingly associated with reduced accumulation of PB and

PC (Supplementary Fig. 10b). These data suggest that IL-10 may be essential for the accumulation of PB and regulatory T cells in lymphoid organs and therefore its absence is associated with increased inflammatory myeloid cells as observed in the IL10KO Rag$^{-/-}$ recipients (Supplementary Figs. 10b, 5c, d). Thus, these studies reveal a previously unidentified role for B cells, specifically PB, in PSA-mediated protection from HSE.

**Discussion**

Encephalitis is a serious, debilitating neuroinflammatory disease resulting from inflammation of the brain caused by an infectious agent or by autoantibodies (autoimmune encephalitis) or by both[33–36]. HSE, the most prevalent viral encephalitis, is associated with high mortality (>20%) and serious neurological sequela despite improvements in diagnosis and antiviral therapy[37–39]. Many studies have identified delayed initiation of iv ACV therapy of >48 h after symptom onset as the most important prognostic indicator of poor patient outcomes[40–43]. Although in mouse model studies, ACV treatment initiated at day 4 pi effectively suppressed HSV replication in the brain and BS by day 6 pi, these ACV-treated mice nonetheless died at the same high rate as control PBS-treated mice and importantly, mortality correlated with massive IM invasion of the BSs of both groups of mice[25]. Delayed ACV treatment lacks protective efficacy because it fails to suppress brain inflammation[26]. Thus, delaying ACV treatment to day 4 pi effectively minimized the contribution of virus-induced pathology, enabling evaluation of PSA as an immunomodulatory agent with the capacity to mitigate uncontrolled CNS inflammation, which is causally involved in fatal HSE.

In sterile inflammatory disease models, PSA-mediated suppression of inflammation depends on its ability to induce regulatory CD4 T cell subsets producing the immunosuppressive cytokine, IL-10. PSA-induced FoxP3$^+$CD4$^+$ T cells, FoxP3$^-$CD4$^+$ Tr1 cells, and CD39$^+$CD4$^+$ T cells all secreted IL-10[13,29,31,44]. Similarly, in our model of viral encephalitis, PSA induced both FoxP3$^+$CD39$^+$Tregs and also FoxP3$^-$CD39$^+$CD73$^+$CD4$^+$ and CD73$^+$CD8$^+$ regulatory T cells. Unexpectedly, Treg depletion in PSA-treated infected WT mice did not impede PSA-mediated protection, suggesting that regulatory CD39$^+$CD73$^+$ T cells were sufficient to compensate and protect from HSE. Moreover, increased numbers of activated CD4 and CD8 T cells secreting both IL-10 and IFNγ were detected in the lymph nodes of PSA-treated mice. Although our results appear antithetical to the concept of a PSA-induced Treg-driven suppression of inflammation, recent studies support our findings. PSA has been shown to activate a TLR2-dependent mechanism to induce CD39$^+$CD4$^+$ T cells that can protect from inflammation similarly to Tregs[31,45]. Furthermore, activated CD45Rb$^{low}$ T effector memory cells cooperated with Tregs to produce maximal levels of IL-10[29]. Altogether, these studies support our finding that PSA induces multiple immunomodulatory cell subsets all of which can secrete IL-10 to suppress inflammation. Thus IL-10, the key mediator of PSA's immunomodulatory effects in various models of sterile inflammation, also functions as the principal effector of PSA-mediated suppression of inflammation in our model of viral encephalitis[11,14,28,29,44,46].

PSA has been shown to bind TLR2$^+$ pDCs and macrophages resulting in IL-10 secretion by regulatory T cells[32,47,48]. Additionally, direct binding of PSA to TLR2 on T cells also induced IL-10 secretion[48]. The c-type lectin receptor, dendritic cell-specific intercellular adhesion molecule-3-grabbing non-integrin (DC-SIGN) was recently proposed as the main receptor for PSA on human DC stimulating CD4 T cell responses[49]. Although PSA binding to pDC and macrophages in gut, but not spleen, was reduced in the absence of SIGNR1, the murine homolog of

DC-SIGN, protection against HSE remained intact. Thus, PSA binding by SIGNR1 is not essential for generation of protective regulatory T cell responses.

Recently, it was reported that macrophages and a subset of B cells bound metabolically labeled *B. fragilis* and PSA[50]. We extended these observations here to show that PB, in addition to pDCs and macrophages in the small intestine, can bind PSA. Importantly, depletion of B cells prior to PSA treatment resulted in a complete loss of IL-10-secreting T cells and in protection from HSE, highlighting the key role of B cells in induction of IL-10-secreting regulatory T cells. Intriguingly, this B cell mechanism appears to be partially IL-10 dependent, since WT but not IL10KO B cells induced complete protection from encephalitis.

B cells are renowned for secreting copious amounts of antibody to protect from infections and to activate CD4 T cell responses. However, their role as a regulatory cell type is now gaining prominence in mainstream immunology. Among B cells, B regulatory cells (Bregs) have received most attention as a regulatory cell. In the gut, Bregs secreting IL-10 were shown to be induced by the microbiota via an IL-6-dependent and IL-1β-dependent differentiation pathway[51]. PB and/or PC secrete antibodies and additionally other factors, including cytokines that can modulate the immune response. In response to microbes in the gut, IgA+ PC have been shown to secrete GM-CSF, IL-17A, TGFβ, and retinoic acid, in addition to IL-10 and IL-35[52,53]. IgA+ PC isolated from the lamina propria induced Tregs by secreting TGFβ and retinoic acid with an important regulatory role for B cells evidenced by the loss of Tregs upon their depletion[54]. Moreover, IL-10-secreting B cells, IgA+ PCs and PB have been shown to be crucial in suppression of inflammation in models of arthritis, colitis, EAE, and chemotherapy, by inducing regulatory T cells and suppressing autoreactive Th1 cells[52,55–57]. CD138+ MHCII+Blimp1+ PC were also shown to be the major producers of IL-10 and IL-35 on TLR4 and CD40 engagement, and these cells were indispensable in the recovery phase in an EAE model[53,58,59]. IL-35 is an immunosuppressive cytokine belonging to the IL-12 family of cytokines that induces the expansion of a subset of Tregs and Bregs, and mediates their suppressive function and inhibits IL-17A[60]. TLR2 and TLR4 stimulation and IRF4 activation were critical for PB secretion of IL-10 in lymph nodes and spleen, which limited inflammation in the CNS and EAE progression[58]. Recently, IL-27 signaling in T cells, promoted by B cells secreting IL-10, was shown to be important for induction of IL-10-secreting Tr1 cells that could suppress Th1-mediated colitis[61]. Combined therapy with donor alloantigens and a probiotic; induced Tregs that conferred enduring protection against GVHD while preserving GVL activity and, similar to our results, this tolerogenic mechanism required IL-10 sufficient B cells in the recipients[62].

The gut environment is conducive to induction of regulatory cells and this is likely why PSA given orally, but not IP or IV, is capable of protecting against HSE. We speculate that PB bind PSA in the small intestine via either TLR2 or TLR4 which is essential for triggering IL-10 production. While we have not examined IL-35 or IL-27 production by PB, they may be secreted in addition to IL-10 and contribute to induction of regulatory T cells. This appears likely, since transfer of IL-10KO B cells into Rag$^{-/-}$ mice increased survival of Rag$^{-/-}$ recipients, though not significantly compared to WT B cell recipients, which suggests that IL-10 from B cells is the critical cytokine for induction of regulatory T cells secreting IL-10. These results suggest that in addition to IL-10, other suppressive cytokines, possibly IL-35 or IL-27, may be required to elicit the full immunomodulatory functions of PSA-activated

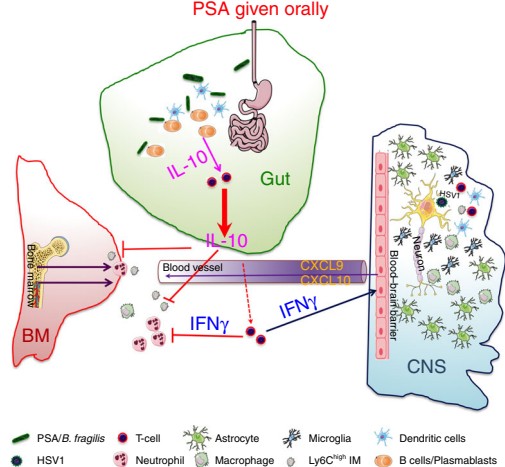

**Fig. 7** Role of the bacterial symbiosis factor PSA in preventing viral encephalitis. HSV infection of susceptible 129 WT mice provokes excessive production of neutrophils (PMN) and Ly6C$^{high}$ inflammatory monocytes (IM) in the bone marrow that invade the brainstem in massive numbers resulting in fatal HSV encephalitis (HSE), despite antiviral treatment from day 4 pi. The bacterial symbiosis factor, PSA given orally is bound by B cells/CD138+ plasmablasts (PB) in the small intestine, which induces IL-10 and IFNγ production by regulatory CD4 and CD8 T cells resulting in the suppression of pathogenic inflammatory myeloid cells concomitant with the induction of IFNγ inducible chemokines in the BS. This novel study reveals the immunomodulatory potential of PSA in protecting from lethal viral infections of the CNS in combination with an antiviral. Cells involved in this protective mechanism are shown in the key. Inhibitory pathways indicated by red blocking arrows

PB to induce regulatory T cells and to protect from viral encephalitis.

This is the first report of PSA or *B. fragilis* exerting potent immunomodulatory activity to protect against a lethal viral neuroinflammatory disease, HSE, that is associated with high mortality and significant morbidity. Our studies reveal a mechanism, whereby PSA delivered orally binds and stimulates intestinal TLR2+ pDCs and B cells to secrete IL-10, followed by induction of regulatory T cells producing IL-10 and IFN-γ that together suppress pathogenic IM and neutrophils to prevent encephalitis[26,27,63], as illustrated in Fig. 7. IFN-γ acting counterintuitively as a regulator of inflammation, suppresses G-CSF and thereby inhibits invasion of the BS by pathogenic neutrophils that are casually involved in HSE[27]. This finding, which we recently reported, explains why IFN-γ is essential for PSA-mediated protection against HSE. Our experimental model emphasizes the protective potential of PSA when given in combination with the ACV, which reveals an interesting synergy as neither PSA nor ACV given alone are protective.

Auto-immune encephalitis triggered by HSE is increasingly emerging as a serious cause of relapse occurring in ~25% of patients in the absence of detectable HSV[64,65]. The disease, characterized by the presence of antibodies to synaptic receptors (such as the NMDA receptor [NMDAR]) and other neuronal surface proteins in serum or CSF from patients is effectively treated with IVIG, a potent immunomodulatory drug for a variety of auto-immune and inflammatory diseases, including HSE as we reported[26]. Thus, we propose that combinatorial ACV + PSA treatment for HSE has significant translational potential, as it might prevent or mitigate development of relapsing auto-immune encephalitis and possibly neurological complications[66,67], both of which are increasingly urgent problems. Combinatorial treatment with PSA, a prototypical bacterial symbiosis factor and an

antiviral may be an effective treatment strategy for viral inflammatory diseases.

## Methods

**Cells and virus**. Mycoplasma-free Vero cells (adult African green monkey kidney cells) obtained from ATCC were cultured in DMEM (Hyclone) supplemented with 10% fetal bovine serum, 10 mM HEPES, and 1% L-glutamine and penicillin–streptomycin at 5% $CO_2$ and 37 °C and were used for virus growth and plaque assays.

Master stocks of herpes simplex virus-1, strain 17+ syn (HSV1) composed only of cell released virus were prepared from mycoplasma-free monolayers of Vero cells. Virus titers were determined by plaque assays using Vero cell monolayers and single use aliquots of virus in Hanks balanced salt solution supplemented with 2% fetal bovine serum were stored at −80 °C. Virus titers in Tg and BS homogenates were determined by plaque assay on Vero cell monolayers[26].

**Bacterial strains and culture conditions**. *B. fragilis* strain NCTC9343 was obtained from the American Type Culture Collection. Bacteria were grown in a 16 l fermenter either in brain heart infusion (BHI) broth (BD Biosciences) or a minimum medium containing 8 g glucose, 1% FBS, 0.5 µg hemin, and 0.5 µg/ml vitamin K in 1 l of RPMI (Invitrogen).

PSA was purified, and quality controlled before being used in these studies. Briefly, *B. fragilis* was grown in a 16 l fermenter, bacterial pellets harvested, and soluble material isolated by phenol/chloroform extraction. Nucleic acids and proteins were digested with DNase/RNase and Pronase K, respectively. PSA was purified using column chromatography. PSA was subjected to 1D NMR analysis and the batch used in this project was deemed pure and of correct chemical structure. The biological activity of PSA was tested using mouse immune cells which were analyzed for IL-10 production during in vitro cultures using ELISA and flow cytometry.

**PSA conjugation with a fluorescent label**. Conjugation of PSA to Alexa Fluor 488 was done according to a method taken from Tranatafilou et al.[68]. Briefly, 1 mg of PSA (lyophilized powder from PBS) in 500 µl $H_2O$ was oxidized with 14.3 µl, 20 mM $NaIO_4$, pH 7.1, for 30 min on ice in the dark. Excess $NaIO_4$ was removed by desalting on a spin column (Zeba, 7KDa MW cutoff, Thermo Scientific), equilibrated with PBS. Alexa 488 hydrazide (14.3 µl, 20 mM) was added to the elutriate, the pH adjusted to 6.2 with dilute HCl and reacted for 2 h at RT in the dark. $NaCNBH_3$ (14.3 µl, 200 mM) was added to and reacted for 2 h at RT in the dark. Excess Alexa 488 hydrazide and $NaCNBH_3$ were removed by desalting on a spin column (Zeba, 7KDa MW cutoff, Thermo Scientific) equilibrated with PBS.

**Mice**. 129 WT (129S6/SvEvTac) and $Rag2^{-/-}$ (129S6/SvEvTac-Rag2tm1Fwa) mice were obtained from Taconic (Hudson, NY), while 129 IL-10KO (129(B6)-IL10tm1Cgn/J) mice backcrossed at least eight times to 129S6/SvEvTac mice were obtained from Jackson Laboratories (Bar harbor, Maine). 129 WT, 129 GKO, 129 $Rag^{-/-}$, and 129 IL-10KO mice were bred in the Animal Research Facility at City of Hope. SIGNR1KO mice backcrossed seven times to BALB/c and their WT littermate controls were obtained from Dr. Andrew McKenzie, MRC laboratory of Molecular Biology, Cambridge, UK[69]. The use of BALB/c mice did not influence the outcome of experiments investigating the role of SIGN-R1 as both strains are susceptible to HSE[70]. Both male and female mice, enrolled in the study at 6–10 weeks of age, were group housed separately at four mice per cage in sterile disposable Innovive cages using the Innorack caging system with dual HEPA filtered transversal air flow ventilation (Innovive), sterile Aquavive water (Innovive) and irradiated Picolab rodent diet 20 (LabDiet); littermates of the same sex were randomly assigned to experimental groups.

**Mouse infection and treatments**. Mice infected with HSV ($10 \times LD_{50}$ for 129 strain: 3200 PFU, BALB/c: $8 \times 10^4$ PFU) via corneal scarification were monitored daily for signs of encephalitis. *B. fragilis* ($10^9$ CFU) or its capsular PSA (2 mg/kg) was administered by oral gavage. The anti-viral drug ACV (50 mg/kg) was given by i.p. injections from day 4 pi to eradicate infectious virus both systemically and in the CNS.

**Adoptive transfers and cell depletions**. $CD3^+$ T cells or $CD19^+$ B cells isolated from WT or IL-10KO mice were cross matched and adoptively transferred by iv injections into 129 Rag mice such that four groups of $n = 10$ mice received all possible cell subset combinations. T cells and B cells were isolated using EasySep mouse CD4, CD8 or CD3 T cell and CD19 B cell immunomagnetic negative selection isolation kits by as per manufacturers recommendations (Stemcell Technologies). B cells in naïve WT mice were depleted by ip injection of 250 µg of αCD20 monoclonal antibody (Genentech Inc.) every 10 days; the absence of $CD19^+$ B cells in the spleen and LN was confirmed by flow cytometry. We confirmed the presence of T cells and absence of B-1 cells using a CD5 marker. $CD25^+$ $FoxP3^+$ Tregs were depleted by ip injection of 250 µg of αCD25 monoclonal antibody (Clone PC61, ATCC) given every 3 days and the absence of $FoxP3^+$ Tregs was confirmed by flow cytometry. $PDCA1^+$ pDCs were depleted by ip injection of

200 µg of αPDCA1 mAb (clone 297, a gift from Dr. Marco Colonna, Washington University School of Medicine) given every 3 days. All depletions were initiated 10 days prior to PSA treatment and continued for the duration of the experiment.

**Isolation of mononuclear cells from the CNS, lymphoid organs, and gut**. Mononuclear cells from spleen, CLN, MLN, blood, and BS were isolated and used in these studies[26]. Briefly, two or three pooled BS were minced and digested with collagenase and DNAse for 30 min at 37 °C after which the cell suspension was centrifuged through a two-step Percoll gradient at 800×g for 25 min at 4 °C. The resulting enriched population of viable mononuclear cells included $CD45^{high}$ leukocytes, $CD45^{int}$ microglia, and $CD45^{neg}$ BS resident glial cells. Cell viability was >90% as revealed by trypan blue staining. Cells from lymph nodes were minced and washed prior to resuspension in RPMI−1640 with 10% fetal calf serum while splenocytes were prepared in this medium following mincing and RBC lysis. Lamina propria, Peyer's patches, and gut IEC were isolated from duodenum, ileum, or colon. Intestinal tissues were collected and washed with ice cold PBS several times to remove the feces. Residual fat was removed, and intestine was cut first longitudinally and then laterally into 1 cm pieces, and placed in 50 ml falcon tubes containing 20 ml of RPMI with 10% FBS, 5 mM EDTA and 1 mM DTT. Tubes were placed on orbital shaker in 37 °C for 30 min. Then supernatant containing epithelial cells and intraepithelial leukocytes was collected, filtered on 100 µm cell strainer, spun down and washed with cold PBS, and after resuspending in cold PBS used for further analysis. Remaining lamina propria tissue was washed with HBSS. Next, tissue was digested using a gentleMacs Octo Dissociator and dissociation kit following the manufacture's protocol (Miltenyi Biotec) prior to straining, centrifugation and resuspension in RPMI medium with 10% fetal calf serum.

**Flow cytometric analysis**. To determine cell surface expression, Ab-labeled cells were acquired on a BD Fortessa Analyzer (BD Biosciences, San Jose, CA) and flow cytometry analysis was performed using FlowJo software (Tree Star Inc.). Doublets were excluded from live cell populations. The gating strategy is shown in Supplementary Fig. 2a–c. CD45 was used to distinguish BM-derived $CD45^{high}$ leukocytes from $CD45^{int}$ $CD11b^+$ microglia and $CD45^{neg}$ neural/glial cells. Neutrophils were determined by their $SSC^{high}$, $Ly6G^+$, $CD115^-$ MHC II⁻ phenotype. $CD4^+$ Tregs were determined by reactivity to CD25 and intracellular FoxP3 expression. Monocytes/macrophages were determined by a $SSC^{low}$ $CD115^+$ $CD11b^+$ $F480^{+/low}$ $Ly6G^-$ phenotype, whereas IMs expressed high levels of Ly6C molecules. Intracellular staining was performed as previously described[27]. Briefly, $10^6$ cells were stimulated for 4–6 h with or without peptide (CD8: $gB_{498–505}$; CD4: heat killed-HSV; or PMA + ionomycin (Sigma-Aldrich), or 100 µg PSA (5 h or overnight) or 500 ng/ml TLR2 agonist LTA-SA (InvivoGen) in the presence of protein transport inhibitors containing Brefeldin A and monensin (eBiosciences). Following FcR blocking, surface expression, using lineage-specific antibodies, was determined. Then, the cells underwent fixation and membrane permeabilization using ebioscience IC fixation/permeabilization buffers (ebioscience), and the permeabilized cells were probed with anti-cytokine Abs to detect cytokines.

The following anti-mouse antibodies obtained from eBioscience, BD Pharmingen, or Biolegend were used in the study: PerCP-Cy5.5-conjugated CD45 (1:100, eBioscience #45-0451-82 or 1:100 Biolegend #103132), eFluor 450-conjugated CD11b (1:100, eBioscience #48-0112-82), BV510-conjugated CD11b (1:150, Bioelegend #101245), PE-conjugated Ly6G (1:200, Biolegend #127607), FITC-conjugated Ly6C (1:100, BD Pharmingen #553104), APC-conjugated Gr-1 (1:100, eBioscience #17-5931-82), PerCP Cy5.5-conjugated CD115 (1:100, Biolegend #135526), APC-conjugated F4/80 (eBioscience #17-4801-82), APC-conjugated MHC II (1:100, eBioscience #17-5321-82), PE-conjugated MHC II (1:200, eBioscience #12-5321-82), PE-conjugated TLR2 (1:150, eBioscience #12-9021-82), PE-conjugated TLR4 (1:150, Biolegend #145403), APC-conjugated SIGNR1 (1:100, eBioscience #17-2093-82), PE-conjugated PDCA1 (1:150, Biolegend #127104), APC-conjugated CD11c (1:100, Biolegend #117310), BV510-conjugated CD11c (1:150, Biolegend #117337), Alexa fluor 700-conjugated CD4 (1:100, Biolegend #100430), BV421-conjugated CD25 (1:100, Biolegend #102043), efluor 450-conjugated CD8a (1:100, eBioscience #48-0081-82), FITC-conjugated CD107a (1:100, eBioscience # 53-1071-82), FITC-conjugated CD107b, (1:100, eBioscience #11-1072-85), PE-conjugated CD103 (1:150, Biolegend #121406), PerCP 710-conjugated ICOS (1:100, eBioscience #46-9940-82), APC-conjugated FoxP3 (1:100, eBioscience #17-5773-82), PE-conjugated CD39 (1:150, eBioscience #12-0391-82), PE-conjugated CD73 (1:150, eBioscience #127206), APC-conjugated CD62L (1:100, eBioscience # 17-0621-82), PerCP Cy5.5-conjugated CD44 (1:100, eBioscience #45-0441-82), FITC-conjugated CD11a (1:100, Biolegend #101006), APC-conjugated MHC class I H-2K$^b$ HSV $gB_{498–505}$ Tetramer (1:400, NIH tetramer facility), APC-conjugated B220 (1:150, eBioscience #17-0452-82), FITC-conjugated CD19 (1:100, Biolegend #115506), PE-conjugated CD138 (1:100, BD Pharmingen #553714), PE-conjugated CD5 (1:150, eBioscience #12-0051-82), APC-conjugated IL-10 (1:100, eBioscience #17-7101-82), PE-conjugated IFNγ (1:150, eBioscience #12-7311-82), efluor 450-conjugated IL-17a (1:100, eBioscience #48-7177-82).

**Statistical analysis**. Graph Pad Prism Software was used to analyze mortality data by the log rank (Mantel–Cox) test, considering both the time of death and

mortality rates. Statistical differences between groups of mice were calculated using two-way ANOVA with Sidaks multiple comparisons tests or ordinary one-way ANOVA with Turkeys multiple comparisons tests to determine the effects of time on cell populations and infiltrations or Student's $t$-tests for other calculations, with $p \leq 0.05$ considered significant in the GraphPad Prism 7 software. The values for $n$ and the definition for $n$ (number of mice) is provided in the relevant figure legends and "Methods" section. All data show mean ± SEM. No data were excluded from this study.

**Ethical statement**. All animal experiments in this manuscript were conducted in strict compliance with the ethical guidelines of the ILAR Guide for the Care and Use of Laboratory Animals and all animal procedures were performed with the approval of the City of Hope Institutional Animal Care and Use Committee.

## Data availability

All relevant data are available upon reasonable request. The source data underlying Figs. 1b, c, 2a–f, 3a–f, 4a–f, 5b, c, 6e and Supplementary Figs. 1a–f, 3e, 4a, b, 4d, 5c, 9a, e are provided in the Source Data file.

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

## Acknowledgements

We thank Dr. Andrew McKenzie, (MRC Laboratory of Molecular Biology, Cambridge, England) for providing BALB/c SIGNR1 KO and control wild type littermate mice. We thank Stacee Mendonca for excellent technical assistance. We thank Dr. Balfour Sartor (University of North Carolina) for providing the 129 IL-10KO spleen cells and Dr. Andrew Chan (Genentech Inc.) for providing anti CD20 monoclonal antibody. Portions of the graphics in Figs. 5 and 7 were obtained from Servier Medical Art, whom we gratefully acknowledge (http://smart.servier.com/). This study was funded by a grant from the Caltech-COH Biomedical Research Initiative to E.M.C. and S.K.M. and by a COH Shared Resources Pilot Program award to C.R.

## Author contributions

E.M.C., S.K.M. and C.R. conceived the project and planned experiments, E.M.C. and S.K.M. provided reagents, H.C. purified PSA, C.R. performed experiments, M.K. isolated gut immune cells, L.L. labeled PSA with Alexa Fluor 488 and E.M.C. and C.R. wrote the paper.

## Additional information

**Competing interests:** E.M.C., C.R. and S.K.M. are inventors of patent application: PCT/US2016/036803, which describes use of PSA as a treatment for viral inflammatory diseases. The remaining authors declare no competing interests.

