## [Peer Review File · Nature Communications]

Reviewers' comments:

Reviewer #1 (Remarks to the Author):

The authors of the manuscript "Bacteroides fragilis Polysaccharide A Induces IL-10 Secreting B and T cells that Prevent Viral Encephalitis" identify a commensal microbe and specifically, its capsular polysaccharide A (PSA), which when given prophylactically can protect against a murine model of fatal viral encephalitis. The authors establish that protection conferred by this microbe requires pretreatment of the mice and results in reduced innate immune inflammatory infiltrates in the central nervous system. Furthermore, they provide mechanistic evidence that this protection depends on B cell and T cell liberated IL-10 and IFN γ . The manuscript is well written and addresses a very relevant area of research that is of broad interest- the identification of specific immunomodulatory molecules from commensal microbes and their mechanism of action. Furthermore, the data regarding protection in this disease model by PSA is very convincing, as pretreatment with *B. fragilis* or PSA increases survival from 0% to greater than 75%. However, there is some reduction in novelty considering that the immunomodulatory functions of PSA were previously known and that IL-10 dependent anti-inflammatory effects of PSA in the central nervous system have previously been extensively described. The difference to previous work in this area is that it involves a viral induced inflammatory condition rather than a sterile inflammation model. Much of the novelty of this manuscript also relies on the implication of immunoregulatory B cells and IL-10 from these cells in the mechanism of reduction of neuroinflammation. While the link to B cells is new with respect to PSA, regulatory B cells and their interplay with regulatory T cells have been described in both sterile (EAE, stroke) and viral (MCMV) neuroinflammatory disease models. In addition, there are several issues in the publication that should be addressed, including concerns over several conclusions not well supported by the data.

Major Concerns

1. In both line 80 and line 90, the authors state that this study addresses the role of the microbiota or that their data demonstrates the importance of the microbiota for optimal antiviral immunity. The data presented do not support these conclusions. In order for this to be true, an analysis of microbiota depletion or dysbiosis (i.e. antibiotics treatment, germ free mice, fecal microbiota transfers) and its effect on the disease model would be required. Instead they present a potentially therapeutic microbe that when given exogenously can protect in this disease model.
2. The gating strategy for B cells as depicted in Figure S2 reveals that B cells are defined as only B220 $^{+}$. However, not just B cells but also pDCs are B220 $^{+}$ cells present in both the MLN and spleen. Since pDCs are known to be important for PSA function and have been shown to be increased in the context of inflammation after PSA treatment (see citation 31 in this publication) it is important to use a gating strategy that distinguishes pDCs and B cells. The inclusion of CD19 $^{+}$ as a marker for B cells would thus be important, especially since the increase in B cells and flow cytometric analysis of B cells is crucial to the main message of this publication.
3. With respect to figure 4A-B, the authors show clearly that PSA induces IL-10 and IFN γ secreting CD4 $^{+}$ T cells, CD8 $^{+}$ T cells and B cells in the MLN and spleen. While it is logical to look at these cells in the MLN, which is the draining lymph node for the intestine where PSA is administered, the connection between why they look only at the MLN and not at the cervical lymph node during neuroinflammation is not clear. The effect of PSA on T and B cell cytokine production in the lymph nodes that actually drain the site of infection should be included, especially since the authors clearly show an increase in CD8 $^{+}$ T cells in the brainstem during infection of PSA treated animals.
4. The authors identify increased virus specific CD8 T cells and plasma cells in the intestines of PSA treated mice. As a result, it would be important to also provide data demonstrating viral load within various organs, including in the intestine to indicate whether this virus disseminates systemically and whether PSA treatment might influence systemic viral load, such as in the intestine.
5. In line 202, the authors draw a conclusion not supported by the data to state that PSA treatment resulted in emigration of B and T cells from the MLN to the spleen. While they do indeed show increased B and T cells in the MLN and the spleen (Figure 3H, 4A-B), they do not provide any

evidence that the source of these cells is migration from the MLN to the spleen.

6. The evidence for increased IL-10 and IFN γ secreting B and T cells in the MLN and spleen following PSA treatment in the context of infection is well demonstrated (Figure 4A-B), however, it would be important to also show whether these cells are increased by PSA treatment in uninfected mice. This is especially important since the message the authors seem to be trying to convey (see line 88-89, 431-434) is that PSA/B. fragilis are optimally priming T cells and the immune response to respond upon subsequent virus infection and thus would be present in uninfected mice.

7. In survival studies involving different knockouts and adoptive transfers (Figure 1B RAG KO, 4E, 5A), the survival data for PBS mock gavaged groups for each genotype/adoptive transfer combination should be included, at least in the supplementary data. As presented it is impossible to interpret what is the true effect of PSA treatment on these groups since they might each have a different baseline response to infection in the absence of PSA treatment.

8. In line 274, the title of the section represents a conclusion not supported by the data. While the authors do demonstrate pDC and plasmablast binding to PSA, they do not link this binding to IL-10 production and protection from HSE. Instead they show just general B cell requirements for protection using B cell depletion, but this is (1) depletion of all B cells and not specifically plasmablasts and (2) no evidence is provided that the plasmablasts bound specifically to PSA have different properties than the non-PSA binding plasmablasts in terms of their ability to stimulate IL-10. In order to directly connect plasmablasts that bind PSA to IL-10 production, additional ex vivo experiments would be required. To directly link plasmablasts to PSA protection, adoptive transfer specifically of plasmablasts to B cell depleted mice and seeing whether it rescues PSA protection would also be revealing. As for pDCs, no link to IL-10 production or protection in this model is provided. Additional experimentation such as using the well established pDC depletion model of BDCA2-DTR mice (which has even been previously used in the context of PSA/neuroinflammation) might strengthen the authors conclusion of pDC involvement.

9. It is unclear why the authors proceed to look at IL-17A and ILC-3 cells in figure 6F. They should address whether IL-17A and ILC-3 cells are involved in HSE pathology and if this is a conclusion they would like to highlight, then IL-17A $^{+}$ T cell levels, in addition to IL-10 and IFN γ , should be included in the previous figures describing the effects of PSA treatment on T and B cell responses during HSE. If unrelated to the story presented here, this data would be better placed in the supplementary information.

10. The authors clearly explain why they chose to look at SIGIRR, but it is unclear why they did not look at the role of TLR2 binding by PSA in this model, since it is well established that PSA binding to TLR2 is important for its immunoregulatory effects and TLR2 deficient mice are readily available from commercial sources. This is especially important since the authors state specifically (line 299) that in the absence of SIGIRR, pDC binding of PSA is "negligible," which would suggest that this is the only mechanism of pDC binding to PSA. Since the publication by Dasgupta et al. clearly demonstrates TLR2 dependent PSA activity by pDCs, it would be important to include a comparison of TLR2 KO and SIGIRR KO PSA binding to address any possible discrepancies with previous work.

Minor concerns

1. In lines 36, 435 and 438, the word prebiotic is misused. A prebiotic is by definition a compound that enhances the growth or activity of a beneficial microbe and has nothing to do with a microbe given prophylactically.

2. With respect to presentation of flow cytometry data, the authors show in figure 3J an increase in total number of T cell subsets in the MLN and use this data to conclude that PSA increases specifically T cells with a regulatory phenotype. A better representation would be to show the T cell populations as percentages of CD4 $^{+}$ so that the reader can see instantly a specific increase in these subsets and not just a reflection of a general increase in all T cell numbers.

3. In line 138-140, the authors state that PSA had no effect on inflammatory infiltrates in the brain stem of Rag KO mice. However, in figure 2A and 2B it does look like there is a trend towards reduced infiltrates in Rag KO mice treated with PSA. Some indication that statistical analyses were performed on these groups would be useful, such as an indication of "ns" for non-significant to show that there really is no significant change.

4. Based on the data presented in figures 3A, 3B, 3H, and 3J, conclusions are made that PSA alters the percentages or number of the relevant cell types in each figure, however no statistical analysis is presented. The authors should add evidence of statistical analysis including “ns” if the findings are found to be not significant and should include reference to the lack of statistical significance in the text.
5. In line 165 of the text the authors state that PSA treatment substantially increased the CD4+ and CD8+ T cells in the MLN, however, based on figure 3I, there is no change in these populations. This statement needs to be corrected to accurately reflect the data.
6. Since the authors have the ability to identify antigen specific T cell subsets (as demonstrated in figure S3B), it would be useful to know if they looked at what percentage of the increased IL-10 and IFN γ secreting T cells were antigen specific and also how PSA altered the cytokine production profile specifically within antigen-specific cells.
7. Line 209 incorrectly states that all of the mice in figures 4C and 4D succumbed to infection, whereas in the context of IL-10KO there appears to have been ~20% survival.
8. In line 291-292, the authors should indicate which organ they are comparing to the gut.
9. In figures S3, S5E, and figure 6, rather than showing a representative flow plot from one mouse, the data would be better presented using some sort of bar graph depicting averages and statistics for multiple mice. It is difficult for the reader to draw any conclusions based on one flow plot.
10. The data regarding SIGNR1 binding, since it is found to have no relation to the story as presented and the protection of HSE would be better placed in the supplementary information.

Reviewer #2 (Remarks to the Author):

This manuscript by Ramakrishna and coworkers deals with an interesting and up to date topic, i.e. the role of microbiota and of bacterial origin factors (the *Bacteroides fragilis* polysaccharide A in this manuscript) in preventing the onset of different diseases (here HSV encephalitis). *Bacteroides fragilis* PSA is considered the archetypical symbiosis factor, as several studies have extensively addressed its role in the prevention/treatment of different disorders, sterile neuroinflammatory conditions included. Thus, the novelty of this manuscript mainly resides: i) in the pathological condition analyzed. Indeed, to my knowledge, there are no previous papers in the literature focusing on the role played by the PSA on HSV encephalitis (HSE) or on viral encephalitis in general; ii) in the observed role of B cells (and specifically intestinal plasmablasts) in mediating the PSA immunomodulatory protective functions in the HSE context. The Authors, starting for the observation that oral PSA administered 3 weeks before viral challenge and during infection/ACV treatment prevents the onset of fatal encephalitis in mice, present several well controlled and robust experiments to illustrate the immunological players involved in this protective anti-inflammatory response. Thus, the main concerns of this Referee, that make me unsure whether this manuscript is suitable for publication in Nature Communications, do not refer to the experimental part of the manuscript, but to the overall relevance of the obtained results in terms of translational potential and of advancements in our understanding of the role played by the microbiota in preventing human diseases, and HSE in particular. Specifically:

1. Although mice represent an accepted model for HSE and the Authors have clearly demonstrated in previous published works the role of neuroinflammation in the HSE pathogenesis in this model, it should be kept in mind that, in the case of the human disease, the relative contribution of viral replication, neuronal cell death, and inflammatory infiltrates to pathology or defense is not entirely clear. Since this aspect is crucial for the general conclusions of this manuscript and its translational potential, it needs to be discussed by the Authors. Furthermore, it would be interesting to know whether there are evidence in the literature of any connection, for instance, between dysbiosis and onset/prognosis of HSE or of other non sterile neuroinflammatory diseases in humans. These or other similar data would strengthen the relevance of this manuscript.

2. Although, as stated by the Authors, HSV1 is the leading cause of sporadic encephalitis, different viruses can cause encephalitis. Thus, in the opinion of this Referee, it should be clearly stated that this manuscript is addressing HSE and not, in general, viral encephalitis, that might be characterized by different pathogenic mechanisms, with diverse levels of neuroinflammation involvement. Thus, for instance, the title should be changed accordingly, as well as additional sentences throughout the text.

3. In the opinion of this Referee, one of the main weakness of this manuscript is represented by the fact that PSA needs to be preventively administered (21 days before HSV challenge) to mice to display an effect on HSE outcome. On the other hand, it was demonstrated that PSA protects mice against allergic encephalomyelitis (EAE) prophylactic and therapeutically (Ochoa-Repáraz et al., *Mucosal Immunol.* 2010 Sep; 3(5): 487-95). These results on one hand underline the complexity of HSE pathogenesis, compared to sterile neuroinflammatory diseases, at least EAE. On the other hand, they significantly limit the translational applications of PSA to HSE in humans, specifically to its treatment. These aspects should be clearly discussed by the Authors. In particular, it should be clearly mentioned that the results of this study support, once again, the role of the microbiota, and of PSA and of *Bacteroides fragilis* in particular, in preventing neuroinflammatory diseases, extending this function to a non-sterile condition. However, the results do not support a role for PSA or for *Bacteroides fragilis* in the treatment of HSE or in general of viral-induced inflammatory diseases. Thus, the Authors should change accordingly some of their conclusions, as for instance last sentence of the Abstract, last sentence of the Introduction, last paragraph of the Discussion. Otherwise, the Authors should better explain how they envisage a "prophylactic" use of PSA for HSE in humans (are there specific at risk populations?).

4. At page 20, in the Discussion section, the Authors state that "neither PSA, nor ACV given alone are protective". However, while it is clearly shown by the experiments reported in the manuscript that ACV alone given 4 days post infection is not protective, this Referee could not find data supporting the statement that PSA alone is not protective. However, this is an important point that should be reported in the Results section and/or better highlighted. Indeed, it is, once again, relevant for the translational implications of this study.

5. Line 108, I would not define the adopted experimental setting "clinically" relevant. Rather, I would state here, as in other parts of the manuscript, that this setting allows to focus on the effect of PSA on the neuroinflammatory part of the HSE pathogenesis in mice. Same thing in the Abstract (line 30). I would remove "to simulate a clinical setting".

Furthermore:

6. There are no comments on the different outcomes of PSA administration via intraperitoneal/ intravenous routes with respect to the oral route. Indeed, only when orally administered PSA is active. This is a relevant result, supporting what is reported at the end of the manuscript, i.e. the importance of the PSA binding to specific gut cells in determining the protective role of this bacterial factor towards HSE. Thus, the Authors should better comment these differences. Also, while at page 10 lines 205-215, data are reported showing the requirement for gamma interferon along with IL-10 for the suppression of infiltrate accumulation in the brainstem of mice, there are no comments in the Discussion on the role played, if any, by gamma IFN. Please comment on it.

7. This manuscript is extremely complex especially for Readers that are not entirely familiar with immunological aspects. In general, some parts of the results section are unclear and difficult to follow as they report, for instance, several references to previous figures or to supplementary data. As an example, see the section entitled "Plasmacytoid DCs and Plasmablasts bind PSA...", where also the repetition of terms like "Importantly", "Interestingly" renders the text a bit confusing. Furthermore, at page 9, line 185-186, the Authors state " Interestingly, PSA treatment increased IL-10 and gamma IFN expression...", few lines below (195-197), the Authors state again "Since we established....we next explored whether PSA induces IL-10 gamma IFN production". Something is missing here to distinguished data reported in the first part of the paragraph from the ones described few lines below. Again, at page 13 lines 276-281, the Authors state why they want to investigate PSA binding to different immune cell population in the intestine of WT and Rag-/- mice (by the way the Authors should always refer to the latter in the same way throughout the text and not as Rag -/- somewhere and as Rag somewhere else; this is confusing). However, first of all it is not clear what the Authors mean by "contrasting outcomes of HSV infection" (line 279).

Secondly, there is no further mention or comment on whether the obtained results answered the question set by the Authors here. In general, the Authors should make an effort in improving the readability of the manuscript. Under this respect, legends to the figures need to be deeply checked, as there are mistakes (for instance punctuation mistakes) that render difficult to understand what is reported in these very complicated, multi-panel figures. For instance, for this Referee it was very hard to understand what the Authors meant in the Legend to Figure 3G when they stated "Treg depletion over 21 days before, during infection and ACV treatment of PSA treated WT mice....". Only going to the Methods and reading how the experiments was performed the sentence and, thus, the figure became clear. Same thing applies to other legends. For instance: legend to Figure 2 "ns" is not defined; legend to figure 3, exchange the comma at line 788 before % with a ";", as it is at line 791; legend to figure 6, remove the full stop after Duod at line 825, remove the comma after (C) at line 828, at line 830 move (D) and (E) after Duod and Ile respectively, MFI at line 788 should be spelled out. Figures themselves need to be carefully checked. As an example, I would suggest to change the color of the circles employed in middle right panels of Figure 5 (B and C) from brown to any other color. Indeed, this might create confusion with the brown circle employed to mark cells reported in the B panels and highlighted also in the legend of the Figure. Title of Figure 6 and of Figure S6 are somehow misleading and should be changed/corrected. A schematic representation of the PSA mechanism of action envisaged by the Authors and summarized at the end of the Discussion might help the Readers and should be introduced as additional Figure. As there are several abbreviations through the text, a list of abbreviations might be useful.

8. Most of the experiments are performed with 129 mice. However, the SIGNR1KO mice derive from BALB/c mice. Considering the well-known diverse susceptibility of different mice strains to HSV and to the onset/severity of HSE, a comment should be made by the Authors stating that the results obtained in the experiments aimed at investigating the role of SIGNR1 in PSA functions were not influenced by the different mice strain adopted in this case.

9. In the opinion of this Referee, at page 13, IEL should be spelled out as intra-epithelial leukocytes and not as intra-epithelial lymphocytes

10. Page 3, line 49: the statement "extensively studied and shown...." need some supporting specific references.

11. While defining *Bacteroides fragilis* probiotic is correct, taking into account the debate around the definition of "prebiotic" (Hutkins et al, *Curr Opin Biotechnol.* 2016 Feb;37:1-7), I would avoid to refer to PSA as prebiotic. Please correct throughout the manuscript

12. Page 14 line 291, the sentence "Similar to...." it is not clear. It looks like something is missing. Same thing for the sentence at page 12, line 258 "Since recipients of WT.....". Page 7 a full stop should be added after "diseases". At page 8 the first sentence should start with "Therefore,....". Page 8 line 165, I would remove "of PSA treated mice".

13. At page 1,0 line 206, after IL-10 (IL-10 KO) should be added.

Response to Review 1

We thank the reviewer for a detailed, thoughtful review and constructive suggestions to improve the manuscript. The reviewer is correct that PSA's immunomodulatory role has only been studied in sterile inflammatory models and that this is the first report of its role in protection from viral infection. In addition to an immunomodulatory role for B cells in protection, we show for the first time that B cells in the gut that bind PSA secrete IL-10 and are involved in induction of IL-10 secreting regulatory T cells.

Major concerns.

1. "the authors state that this study addresses the role of the microbiota or that their data demonstrates the importance of the microbiota for optimal antiviral immunity. The data presented do not support these conclusions"

The reviewer is correct. However, it has been reported that antibiotic treated B6 mice died more rapidly from HSV2 mucosal infection than water treated mice¹. In contrast, we found that germ free status or antibiotic treatment of susceptible 129 mice did not impact survival of mice ocularly inoculated with HSV (not shown). Thus, we state instead that our data demonstrate that an immunomodulatory bacterium or a purified bacterial symbiosis factor when given exogenously can modulate host immunity to protect from HSE.

2. "the gating strategy as depicted in Figure S2 reveals that B cells are defined as only B220+.."

We had used a lymphocytic gate to separate lymphocytes from pDCs, given the relative paucity of DCs in MLN and spleen isolated from naïve mice. We also used CD19 and surface Ig expression to distinguish plasmablasts from B cells. However, we agree that it is better to use a B220+ CD19+ gate to distinguish B cells from pDCs as suggested by the reviewer. We have now changed Fig S2 to reflect this gating strategy and the text (lines 285-287) to reflect CD19 expression by B cells. We used CD19 antibodies to isolate donor B cells for adoptive transfers to Rag mice or anti CD20 antibodies to deplete B cells in wildtype mice, which excludes cross reactivity with or contamination from pDCs.

3. "With respect to figure 4A-B, the authors show clearly that PSA induces IL-10 and IFN γ secreting CD4, CD8 T cells and B cells in the MLN and spleen" and "the connection between why they look only at the MLN and not at the cervical lymph node during neuroinflammation is not clear."

In Figure 4A-B, we showed that PSA increased IL-10 secreting CD4 and CD8 T cells but not B cells in the draining cervical lymph nodes, which was labeled as CLN in the figure and in the figure legend. However, the text referred only to IL-10 and IFN γ production by B and T cells in the MLN; we now state in the text that T, but not B cells, in the CLN produce IL-10 and IFN γ .

4. "The authors identify increased virus specific CD8 T cells and plasma cells in the intestines of PSA treated mice." "...it would be important to also provide data demonstrating viral load within various organs"

All mice used in the different experiments were injected ip with the anti-viral drug acyclovir daily from day 4 pi for a week (Fig. 1), which completely inhibits virus replication in all organs. We added a sentence in the methods section to emphasize this fact.

5. "In line 202, the authors draw a conclusion not supported by the data to state that PSA treatment resulted in emigration of B and T cells from the MLN and spleen.."

On feeding mice with fluorescently conjugated PSA, we did not find PSA labeled cells in the spleen but only in the intestine and after 8-24 hours in the MLN as well; we did not include this data in the manuscript. However, the reviewer is correct in that we have not definitively shown emigration of B and T cells from MLN to spleen, hence, we removed this statement from the results section.

6. "The evidence for increased IL-10 and IFN γ secreting B and T cells in the MLN and spleen following PSA treatment in the context of infection is well demonstrated (Figure 4AB), however, it would be important to also show whether these cells are increased by PSA treatment in uninfected mice."

We have now included data in supplemental figure 4d to show that PSA treatment induced IFN γ but not IL-10 production from T cells in uninfected PSA treated mice and that IL-10 production occurred only after HSV infection. This result suggests that the regulatory T cells are increased in the spleen and MLN by PSA treatment (Fig 3j), but that they secrete IL-10 (Fig 4a) only in response to infection; we updated the results section accordingly.

7. "In survival studies involving different knockouts and adoptive transfers (Figure 1B, 4E, 5A), the survival data for PBS mock gavaged groups for each genotype/adoptive transfer combination should be included, at least in the supplementary data."

129 and BALB/c wildtype and 129-RAG mice do not survive HSV encephalitis and we have published this previously^{2,3}. We added data in Figure 1c and Supplemental figure 1e to show that PBS and ACV treated 129 WT and Rag mice adoptively transferred with various immune cell subsets do not survive encephalitis.

8. "In line 274, the title of the section represents a conclusion not supported by the data. While the authors do demonstrate pDC and plasmablast binding to PSA, they do not link this binding to IL10 production and protection from HSE."

The reviewer is correct in that we did not link PSA binding by plasmablasts (PB) and pDCs to IL10 secretion and protection from encephalitis. However, in new data (Fig 6c, e) included in the revised manuscript, we show that in vitro treatment of intestinal pDCs and B cells with PSA results in IL-10 production. However, since we do not show this effect in vivo or ex vivo, we have modified the section title to state only that pDCs and B cell plasmablasts bind PSA (line 273).

The adoptive transfer experiment suggested by the reviewer to directly link PB to protection is not feasible because PB have a short half-life and are sessile. Additionally, the paucity of PB in naïve mice renders successful adoptive transfers of these subsets very difficult, if not impossible, to perform. Therefore, in the revised paper we emphasize the novel role of B cells, rather than PB, in PSA's anti-inflammatory effects, which is an important new finding.

Our data reveal the critical importance of B cells, but not pDCs, for protection in the HSE model, because anti CD20 antibody depletion of B cells in wildtype mice eliminated protection, despite pDCs being present. Furthermore, adoptive transfers of CD3+ T cells and CD19+ B cells to Rag mice protected them from encephalitis, emphasizing that B cells rather than pDCs are critical for protection in this model. Regarding the use of BDCA2-DTR mice for pDC depletion, this mouse exists only on the resistant C57B6 mouse background and not on the susceptible 129S6 or BALB/c mouse background and, as the C57B6 BDCA1-DTR mice are resistant to HSV encephalitis they cannot be used for this purpose. However, we have included new data in Supplementary Figure 9a which shows that PSA treated pDC depleted mice (using anti PDCA1 antibody from Dr. Marco Colonna) were protected from HSE, similar to control mice. This new data further emphasizes the role of B cells rather than pDCs in providing protection in a non-sterile inflammatory model. Given these issues, we removed “IL10 production” and changed the section heading (line 273) to state that B cells and pDCs in the gut bind PSA.

9. “It is unclear why the authors proceed to look at IL-17A and ILC-3 cells in figure 6F.”

Several studies have used IL-17A as a readout for inflammation in the gut and IL-10 as a protective response following PSA gavage. Consistent with the absence of CD3+ lymphocytes in Rag mice, we found that IL17+ cells mice were not of myeloid origin but rather derived from ILC-3. However, since IL-17 and ILC-3 are not central to this manuscript we have moved these data to Supplemental figure 6d as suggested by the reviewer.

10. “The authors clearly explain why they chose to look at SIGNR1, but it is unclear why they did not look at the role of TLR2 binding by PSA in this model, since it is well established that PSA binding to TLR2 is important for its immunoregulatory effects...”

We did not look at TLR2KO mice since they are readily available only on the resistant C57B6 mouse background. Dasgupta et al (2014) showed that with PSA stimulation, ~5% of pDC (bone marrow derived DC) expressed TLR2 while control media treated bone marrow derived pDC did not express TLR2. Our results show that while PSA binding is reduced in the absence of SIGNR1, ~12-14% of gut derived pDCs in the SIGNR1KO mice still bind PSA, a frequency similar to that shown in Dasgupta’s paper; thus our data do not exclude TLR2 binding PSA on pDC (Supplementary Figure 8, 9b-d). Moreover, we do not see differences in binding of PSA to pDCs in the spleens of WT and SIGNR1KO mice (not shown). However, an important distinction is that Dasgupta et al generated pDC from bone marrow cultures, while we analyzed PSA binding to gut resident pDCs. Another important difference is that all data in the Dasgupta et al paper were generated using C57B6 mice whereas our data were derived using 129S6 and BALB/c mice. Thus, our results cannot be compared directly to those of Dasgupta because of the use of different mouse strains and inflammation models (sterile vs viral), which may account for some of the observed differences, including importantly that in addition to pDCs and macrophages, we show for the first time that gut resident B cells bind PSA and produce IL-10 but only after infection. Despite these differences our results do not contradict results from Dasgupta et al⁴ obtained in a different model.

Minor concerns.

1. "In lines 36, 435 and 438, the word probiotic is misused."

We agree and thank the reviewer for bringing this important point to our attention; in the revised paper we describe PSA as a 'symbiosis factor'.

2. "With respect to presentation of flow cytometry data,."

In our experience in our field of study it is typical to regard an increase in numbers of T cells as reflective of function. An increase in numbers of regulatory T cells as shown in Figure 3J demonstrates that PSA induces regulatory T cells and usually, reviewers request absolute cell numbers rather than just the percentage increase.

3. In line 138-140, the authors state that PSA had no effect on inflammatory infiltrates.."

There was no statistical difference in infiltration in BS of Rag mice and this is indicated as 'ns' for 'not significant' in figure 2A and 2B.

4. "Based on data presented in figure 3A, 3B, 3H and 3J..."

Statistics added for figures 3A, 3B, 3H and 3J.

5. "In line 165 of the text the authors state that PSA treatment substantially increased the CD4+ and CD8+ T cells in the MLN ..."

Figure 3H shows that PSA increases CD4 and CD8 T cell accumulation in the MLN. Increased cellularity in lymph nodes is demonstrated by an increase in numbers as shown in 3J. Figure 3I shows percentage composition which typically does not change much as lymph nodes and spleen are composed of defined percentages of T cells and B cells, with only small variations in percentages but large changes in absolute cell numbers during microbial infections.

6. "Since the authors have the ability to identify antigen specific T cell subsets (as demonstrated in figure S3B),"

We have not looked at cytokine production in tetramer positive cells, as in our experience, surface tetramer expression is down regulated following cognate antigen stimulation to detect intracellular cytokine expression.

7. "Line 209 incorrectly states that all the mice in figures 4C and 4D succumbed to infection..."

We thank the reviewer for pointing this out; we have now changed the line to read "the majority of IL10KO succumbed to infection."

8. "In 291-292, the authors should indicate which organ they are comparing to gut."

Clarified as spleen.

9. In figures S3, S5E and figure 6, rather than showing representative flow plot from one mouse, the data is better represented as bar plots with averages..”

We believe FACS plots show differences in binding better. Furthermore, this data shows cells isolated from pooled brainstems or gut cells from one experiment and is representative of 2-3 such experiments. We have now included for Figure 6 bar graphs that summarize data from flow plots.

10. “The data regarding SIGNR1 binding, since it is found to have no relation to the story as presented and the protection of HSE would be better placed in the supplementary information.”

We agree with the reviewer that SIGNR1 does not influence PSA protection from HSE. We thank the reviewer for this insightful suggestion and have now moved all the SIGNR1 data into supplemental information.

Response to Reviewer 2:

We thank this reviewer for insightful and pertinent comments.

1. The main concern of this reviewer relates to their uncertainty as “to the overall relevance of the obtained results in terms of translational potential and of advancements in our understanding of the role played by the microbiota in preventing human diseases, and HSE in particular.”

Most studies of HSV pathogenesis have been done in mouse models and results from these studies have significantly advanced our understanding of the pathologic mechanisms underlying HSE, especially immunopathological mechanisms. We agree that for humans, the relative contribution of viral replication, immune infiltrates and intrinsic cellular mechanisms to HSE remains unclear. However, we proposed combinatorial ACV and PSA treatment as a translational strategy to effectively suppress both virus replication and inflammation mediated pathology. Our results show this is highly effective even when ACV treatment is delayed to day 4 pi, which is clinically relevant as delayed ACV treatment is the most important prognostic indicator of poor patient outcome. See response to point 3 below regarding translational potential.

“Furthermore, it would be interesting to know whether there are evidence in the literature of any connection, for instance, between dysbiosis and onset/prognosis of HSE or of other non-sterile neuroinflammatory diseases in humans.”

There is a report that Hepatitis C virus (HCV) infected, compared to healthy individuals, show gut dysbiosis characterized by changes in abundance of specific bacterial taxa. Alterations in the microbiota were evident even in individuals with mild liver disease⁵. In HIV infected Ugandans, low CD4 T cell numbers have been linked to alterations in the bacterial microbiome as well as to enteric adenovirus expansion⁶. Gut dysbiosis is also seen in HIV and SIV infected patients and rhesus macaques respectively with complex interactions between anti-retroviral therapy and HIV effects on CD4+ T cells causing profound alterations to the microbiota⁷⁻⁹. A very recent paper reports that sub-lethal influenza virus infection results in transient depletion of intestinal bacteria and disruption of

the mucous layer, which fosters increased susceptibility of the small intestine to bacterial pathogen infection¹⁰.

2. “.. in the opinion of this Referee, it should be clearly stated that this manuscript is addressing HSE and not, in general, viral encephalitis, that might be characterized by different pathogenic mechanisms, with diverse levels of neuroinflammation involvement.”

Encephalitis is defined as inflammation of the brain and we agree that different viral infections elicit diverse levels of inflammatory infiltrates. Nonetheless, the immunomodulatory activities of, for example, intravenous immunoglobulin (IVIG) protect against HSV and WNV encephalitis as we showed¹¹, as well as other encephalitides including eastern equine encephalitis (EEE) virus¹² and tick borne encephalitis¹³. As immunomodulatory agents, IVIG and PSA have shown benefit in preventing or moderating a wide spectrum of inflammatory and autoimmune diseases in animal models and patients in the case of IVIG – references in the introduction for PSA and for IVIG in other publications^{3,11,14}. Thus, we respectfully disagree with the reviewer’s contention that our results are not relevant to encephalitis caused by other viruses.

3. “...one of the main weakness of this manuscript is represented by the fact that PSA needs to be preventively administered (21 days before HSV challenge) to mice to display an effect on HSE outcome.”

We used a robust model inoculating mice with a 10x LD50 HSV dose to test protection against HSE in genetically susceptible 129 mice, which results in rapid development of HSE and death of all mice despite ACV treatment from day 4 pi. Disease progression is simply too rapid with the induction of enormous number of inflammatory monocytes and neutrophils and myeloid derived suppressor cells within the first couple days of infection, which effectively suppress generation of protective T cell responses when PSA is given even 24 h pi. This is consistent with studies showing T cell responses developing too late to prevent HSV accessing the brain in wt mice¹⁵. However, the situation in humans is likely different since delayed ACV treatment does not always preclude survival albeit with severe neurological complications. On the other hand, EAE is a slowly progressing disease initiated by small numbers of CD4 T cells that provoke innate cells to cross the blood brain barrier, induce chemokines that permit T cells to traffic into the spinal cord and induce disease. This slowly progressing disease which spans over 9-12 days of “pre-clinical disease” provides an extended window of time for PSA and other such immunomodulatory agents given therapeutically to alter either T cell mediated or innate cell mediated responses.

“ ... the results do not support a role for PSA or for *Bacteroides fragilis* in the treatment of HSE or in general of viral induced inflammatory diseases” and “the Authors should better explain how they envisage a “prophylactic” use of PSA for HSE in humans (are there specific at risk populations?).

Auto-immune encephalitis triggered by HSE is increasingly emerging as a serious cause of relapse occurring in ~25% of patients in the absence of detectable HSV^{16,17}. The disease, characterized by the presence of antibodies to synaptic receptors (such as the NMDA receptor [NMDAR]) and other neuronal surface proteins in serum or CSF from patients is effectively treated with IVIG, a potent immunomodulatory drug for a variety of auto-immune diseases and inflammatory diseases including HSE as we reported¹¹. PSA can prevent and even reverse ongoing experimental

autoimmune encephalomyelitis (EAE), the animal model for multiple sclerosis (MS), an autoimmune disease¹⁸. Notably, naïve CD4+ T cells from MS patients acquire a regulatory phenotype when treated ex vivo with PSA and Tregs are essential for prevention of autoimmunity¹⁹. Thus, we argue that combinatorial ACV + PSA treatment has significant translational potential for the treatment of HSE since it can be expected to prevent or mitigate development of relapsing auto-immune encephalitis, an increasingly urgent problem. Furthermore, we envisage that this strategy could also alleviate development of neurologic complications following HSE, based on recent results suggesting combinatorial ACV+IVIG could be beneficial^{14,20}. We expanded discussion of the translational potential of combinatorial antiviral + PSA treatment for HSE and other viral inflammatory diseases in the revised manuscript (lines 453-464).

4. “neither PSA, nor ACV given alone are protective; this is an important point that should be reported in the Results section’

We stated in results (line 116-117) that PSA nor ACV given alone was not protective and this data is now included in Supplementary Figure 1.

5. “I would not define the adopted experimental setting “clinically” relevant. Rather, I would state here, as in other parts of the manuscript, that this setting allows to focus on the effect of PSA on the neuroinflammatory part of the HSE pathogenesis in mice”.

We agree and have changed the text accordingly.

6. “There are no comments on the different outcomes of PSA administration via intraperitoneal/intravenous routes with respect to the oral route... the Authors should better comment these differences.”

We included a comment on why PSA given orally but not IP or IV is protective against HSE (line 377). We have shown that while DCs isolated from spleen and the gut express both TLR2 and TLR4, B cells in the gut, but not the spleen, express TLR2. This result agrees with several studies which show that mature B cells in the spleen do not express TLR2 or TLR4, but that plasma cells express a whole range of TLRs. Since in our viral inflammatory model, PSA protection is dependent on B cells rather than DCs, it stands to reason that gut resident TLR2⁺ B cells that recognize PSA orchestrate protection by induction of regulatory T cells. Thus the oral route of delivery is important as it allows PSA to access B cells in the gut.

“page 10 lines 205-215, data are reported showing the requirement for gamma interferon along with IL10 for the suppression of infiltrate accumulation in the brainstem of mice, there are no comments in the Discussion on the role played, if any, by gamma IFN. Please comment on it”.

We added a sentence in the discussion (line 429-432) explaining the requirement for IFN γ in PSA mediated protection against HSE. We have shown previously that in the absence of IFN gamma, G-CSF induces pathogenic, apoptosis resistant neutrophils, which invade the brain to cause HSE. We observe a similar anti-inflammatory role for IFN γ in PSA mediated protection and expand on its role in the discussion. his manuscript is extremely complex especially for Readers that are not entirely

familiar with immunological aspects.... the Authors should make an effort in improving the readability of the manuscript. Under this respect, legends to the figures need to be deeply checked, as there are mistakes (for instance punctuation mistakes) that render difficult to understand what is reported in these very complicated, multipanel figures”.

We have attempted to improve readability of the revised manuscript by rewording any sentence that might be difficult to understand, paying particular attention to the figure legends.

7. “A schematic representation of the PSA mechanism of action envisaged by the Authors and summarized at the end of the Discussion might help the Readers and should be introduced as additional Figure”.

We thank the reviewer for this excellent suggestion. We have included an additional figure that summarizes the PSA mechanism of action at the end of the manuscript.

8. “Most of the experiments were performed with 129 mice except for the SIGNR1KO mice that are on the BALB/c background. Considering the well-known diverse susceptibility of different mice strains to HSV and to the onset/severity of HSE, a comment should be made by the Authors stating that the results obtained in the experiments aimed at investigating the role of SIGNR1 in PSA functions were not influenced by the different mice strain adopted in this case”.

BALB/c mice are susceptible to HSE and survive at marginally higher rate than 129 mice. We included a statement in the methods stating that the use of BALB/c mice did not influence the outcome of experiments investigating the role of SIGN-R1 as both strains are susceptible to HSE.

9. “IEL should be spelled out as intraepithelial leukocytes and not as intraepithelial Lymphocytes”

Done.

10. “Page 3, line 49: the statement “extensively studied and shown....” need some supporting specific references.

Done.

11. “While defining *Bacteroides fragilis* probiotic is correct, taking into account the debate around the definition of “prebiotic” (Hutkins et al, Curr Opin Biotechnol. 2016 Feb;37:17), I would avoid to refer to PSA as prebiotic. Please correct throughout the manuscript”.

Thank you for bringing this important issue to our attention. We now refer to PSA as a ‘symbiosis factor’ throughout the revised manuscript.

12. “Page 14 line 291, the sentence “Similar to.....” it is not clear. It looks like something is missing. Same thing for the sentence at page 12, line 258 “Since recipients of WT.....”. Page 7 a full stop should be added after “diseases”. At page 8 the first sentence should start with “Therefore,....”. Page 8 line 165, I would remove “of PSA treated mice”.

We are grateful to the reviewer for exposing errors that confounded upstanding of certain sentences. We modified the offending sentences so that their meaning is now clear.

References:

1. Oh, J.E., *et al.* Dysbiosis-induced IL-33 contributes to impaired antiviral immunity in the genital mucosa. *Proceedings of the National Academy of Sciences* **113**, E762-E771 (2016).
2. Lundberg, P., *et al.* A locus on mouse chromosome 6 that determines resistance to herpes simplex virus also influences reactivation, while an unlinked locus augments resistance of female mice. *J Virol* **77**, 11661-11673 (2003).
3. Ramakrishna, C., *et al.* Establishment of HSV1 Latency in Immunodeficient Mice Facilitates Efficient *In Vivo* Reactivation. *PLoS Pathog* **11**, e1004730 (2015).
4. Dasgupta, S., Erturk-Hasdemir, D., Ochoa-Reparaz, J., Reinecker, H.-C. & Kasper, Dennis L. Plasmacytoid Dendritic Cells Mediate Anti-inflammatory Responses to a Gut Commensal Molecule via Both Innate and Adaptive Mechanisms. *Cell Host & Microbe* **15**, 413-423 (2014).
5. Inoue, T., *et al.* Gut Dysbiosis Associated With Hepatitis C Virus Infection. *Clin Infect Dis* **67**, 869-877 (2018).
6. Monaco, Cynthia L., *et al.* Altered Virome and Bacterial Microbiome in Human Immunodeficiency Virus-Associated Acquired Immunodeficiency Syndrome. *Cell Host & Microbe* **19**, 311-322 (2016).
7. Ji, Y., *et al.* Changes in intestinal microbiota in HIV-1-infected subjects following cART initiation: influence of CD4+ T cell count. *Emerging microbes & infections* **7**, 113-113 (2018).
8. Mudd, J.C. & Brenchley, J.M. Gut Mucosal Barrier Dysfunction, Microbial Dysbiosis, and Their Role in HIV-1 Disease Progression. *The Journal of Infectious Diseases* **214**, S58-S66 (2016).
9. Ortiz, A.M., *et al.* Experimental microbial dysbiosis does not promote disease progression in SIV-infected macaques. *Nature Medicine* **24**, 1313-1316 (2018).
10. Yildiz, S., Mazel-Sanchez, B., Kandasamy, M., Manicassamy, B. & Schmolke, M. Influenza A virus infection impacts systemic microbiota dynamics and causes quantitative enteric dysbiosis. *Microbiome* **6**, 9 (2018).
11. Ramakrishna, C., Newo, A.N.S., Shen, Y.-W. & Cantin, E. Passively Administered Pooled Human Immunoglobulins Exert IL-10 Dependent Anti-Inflammatory Effects that Protect against Fatal HSV Encephalitis. *PLoS Pathog* **7**, e1002071 (2011).
12. Mukerji, S.S., Lam, A.D. & Wilson, M.R. Eastern Equine Encephalitis Treated With Intravenous Immunoglobulins. *The Neurohospitalist* **6**, 29-31 (2016).
13. Ruzek, D., Dobler, G. & Niller, H. May early intervention with high dose intravenous immunoglobulin pose a potentially successful treatment for severe cases of tick-borne encephalitis? *BMC Infectious Diseases* **13**, 306 (2013).
14. Ramakrishna, C., *et al.* Effects of Acyclovir and IVIG on Behavioral Outcomes after HSV1 CNS Infection. *Behavioural Neurology* **2017**, 14 (2017).
15. van Lint, A., *et al.* Herpes Simplex Virus-Specific CD8+ T Cells Can Clear Established Lytic Infections from Skin and Nerves and Can Partially Limit the Early Spread of Virus after Cutaneous Inoculation. *J Immunol* **172**, 392-397 (2004).
16. Nosadini, M., *et al.* Herpes simplex virus-induced anti-N-methyl-d-aspartate receptor encephalitis: a systematic literature review with analysis of 43 cases. *Developmental Medicine & Child Neurology* **59**, 796-805 (2017).

17. Armangue, T., *et al.* Frequency, symptoms, risk factors, and outcomes of autoimmune encephalitis after herpes simplex encephalitis: a prospective observational study and retrospective analysis. *The Lancet Neurology* **17**, 760-772 (2018).
18. Ochoa-Reparaz, J., *et al.* Central Nervous System Demyelinating Disease Protection by the Human Commensal *Bacteroides fragilis* Depends on Polysaccharide A Expression. *The Journal of Immunology* **185**, 4101-4108 (2010).
19. Burgess, J.N., Pant, A.B., Kasper, L.H. & Colpitts Brass, S. CD4+ T cells from multiple sclerosis patients respond to a commensal-derived antigen. *Annals of Clinical and Translational Neurology* **4**, 825-829 (2017).
20. Iro, M.A., *et al.* Immunoglobulin in the Treatment of Encephalitis (IgNiTE): protocol for a multicentre randomised controlled trial. *BMJ Open* **6**(2016).

REVIEWERS' COMMENTS:

Reviewer #1 (Remarks to the Author):

I think the authors have satisfactorily dealt with my concerns. This is a manuscript worthy of publication

Reviewer #2 (Remarks to the Author):

The Authors have significantly improved the quality and readability of their manuscript that is now, in the opinion of this Referee, suitable for publication.